# Autophagy-independent role of ATG9A vesicles as carriers for galectin-9 secretion

Wenting Zhang[1,3], Cuicui Ji[1,3], Xianghua Li[1,3], Tianlong He [1,3], Wei Jiang[1], Yukun Liu[1], Meiling Wu[1], Yunpeng Zhao[1], Xuechai Chen[1], Xiaoli Wang[1], Jian Li[2], Haolin Zhang [1] & Juan Wang [1]

Galectins play vital roles in cellular processes such as adhesion, communication, and survival, yet the mechanisms underlying their unconventional secretion remain poorly understood. This study identifies ATG9A, a core autophagy protein, as a key regulator of galectin-9 secretion via a mechanism independent of classical autophagy, secretory autophagy, or the LC3-dependent extracellular vesicle loading and secretion pathway. ATG9A vesicles function as specialized carriers, with the N-terminus of ATG9A and both carbohydrate recognition domains of galectin-9 being critical for the process. TMED10 mediates the incorporation of galectin-9 into ATG9A vesicles, which then fuse with the plasma membrane via the STX13-SNAP23-VAMP3 SNARE complex. Furthermore, ATG9A regulates the secretion of other proteins, including galectin-4, galectin-8, and annexin A6, but not IL-1β, galectin-3, or FGF2. This mechanism is potentially conserved across other cell types, including monocytic cells, which underscores its broader significance in unconventional protein secretion.

Galectins are characterized by their ability to recognize and bind β-galactosides, enabling them to perform diverse functions such as recognizing damaged intracellular membranes, regulating glycoprotein activity, and acting as a danger signal during bacterial and viral invasions[1-3]. Galectins play pivotal roles in immune response, inflammation, cell migration, signal transduction, cell cycle, cell differentiation, and cell death[4,5]. Despite the broad subcellular localization of galectins, spanning from the cytoplasm to the nucleus[4,5], many of their essential physiological functions are carried out extracellularly as secreted proteins. However, lacking signal peptides, galectins rely on the unconventional protein secretion (UPS) pathway for their extracellular transport[6]. Previous research on galectin secretion primarily focuses on galectin-1 and galectin-3, identifying secretion types such as direct translocation (type I), use of membranous organelles (type III) like exosome-containing multivesicular bodies (MVBs), and secretory autophagosomes[7].

Elevated galectin-9 levels have been associated with immune-related disorders, including autoimmune disorders, viral infections, and parasitic invasion, as well as cancer and various metabolic and cardiovascular diseases[8,9]. Notably, neutralizing anti-galectin-9 antibodies have shown promise in enhancing the efficacy of paclitaxel-based chemotherapy[10]. Despite its potential as a biomarker for disease severity and a therapeutic target[9,11], the mechanism underlying galectin-9's unconventional secretion remains poorly understood.

In our investigation of galectin-9 interacting partners, previous high-throughput mass spectrometry studies have reported an interaction between galectin-9 and the autophagy-related protein ATG9A[12-14]. ATG9, initially characterized as a core regulator of autophagy in Saccharomyces cerevisiae[15], is a transmembrane protein with scramblase activity, capable of translocating phospholipids and facilitating autophagosome formation[16,17]. Mammalian ATG9A vesicles exhibit high mobility, cycling between the trans-Golgi network (TGN), endosomes, plasma membrane, and the pre-autophagosomal

[1]College of Chemistry and Life Science, Beijing University of Technology, Beijing, China. [2]Department of Biochemistry and Molecular Biology, School of Basic Medical Sciences, Shandong University, Jinan, Shandong, China. [3]These authors contributed equally: Wenting Zhang, Cuicui Ji, Xianghua Li, Tianlong He. ✉e-mail: haolin.zhang@bjut.edu.cn; juanwang@bjut.edu.cn

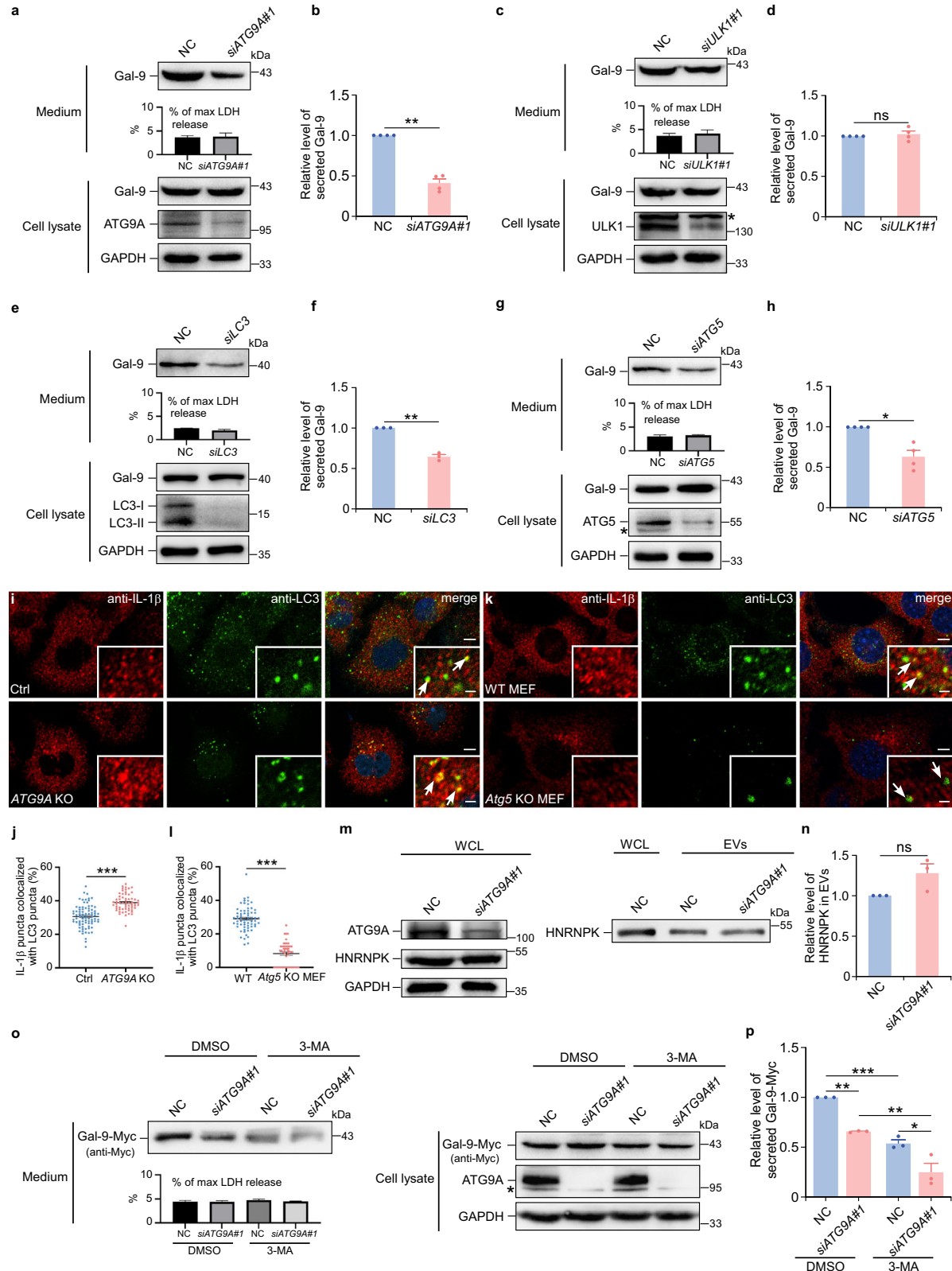

structure[18]. In addition to its classical role in autophagy, ATG9A has been implicated in various non-autophagic functions, including plasma membrane repair[19], lamellipodium expansion[20], autophagy-independent lysosomal targeting pathway[21,22], and lysosomal hydrolase export from the TGN[23].

This study uncovers a critical role for ATG9A in regulating the unconventional secretion of galectin-9. This mechanism operates independently of classical autophagy, autophagy-independent lysosomal targeting, exosome, or LC3-dependent extracellular vesicle loading and secretion pathways. Instead, ATG9A vesicles serve as carriers in a distinct transport system. We also identified the STX13-SNAP23-VAMP3 SNARE complex as a key mediator of ATG9A vesicle fusion with the plasma membrane. Furthermore, ATG9A's regulatory scope extends to other unconventional protein secretion processes,

**Fig. 1 | ATG9A regulates galectin-9 secretion.** Galectin-9 secretion in control (NC) or *ATG9A*-depleted (si*ATG9A*#1) HeLa cells after 24-h DMEM starvation was analyzed by immunoblotting (**a**) and quantified (**b**). Data are presented as mean ± SEM and analyzed using Student's t-test (*n* = 4 biologically independent experiments). **p < 0.01. Galectin-9 secretion in control (NC) or *ULK1*-depleted (si*ULK1*#1) HeLa cells after 24-h DMEM starvation was analyzed by immunoblotting (**c**) and quantified (**d**). Asterisk indicates non-specific bands. Data are presented as mean ± SEM and analyzed using Student's t-test (*n* = 4 biologically independent experiments). ns not significant. Galectin-9 secretion in control (NC) or *LC3*-depleted (si*LC3*) HeLa cells after 24-h DMEM starvation was analyzed by immunoblotting (**e**) and quantified (**f**). Data are presented as mean ± SEM and analyzed using Student's t-test (*n* = 3 biologically independent experiments). **p < 0.01. Galectin-9 secretion in control (NC) or *ATG5*-depleted (si*ATG5*) HeLa cells after 24-h DMEM starvation was analyzed by immunoblotting (**g**) and quantified (**h**). Asterisk indicates non-specific bands. Data are presented as mean ± SEM and analyzed using Student's t-test (*n* = 4 biologically independent experiments). *p < 0.05. **i, j** Immunostaining of endogenous IL-1β and LC3 in control and *ATG9A* KO HeLa cells after 1-h DMEM starvation. **i** Representative immunofluorescence images. **j** Quantification of colocalization. Data are presented as mean ± SEM and analyzed using Welch's t-test

(*n* = 82 in control cells, *n* = 63 in *ATG9A* KO cells examined over 3 independent experiments). Scale bars, 5 μm; inserts, 1 μm. ***p < 0.001. **k, l** Endogenous LC3 immunostaining in MEF control and *Atg5* KO cells expressing IL-1β-FLAG following 1-h DMEM starvation. **k** Immunofluorescence images. **l** Colocalization quantification. Data are presented as mean ± SEM and analyzed using Wilcoxon rank sum test (*n* = 64 in control cells, *n* = 50 in *Atg5* KO MEF cells examined over 3 independent experiments). Scale bars, 5 μm; inserts, 1 μm. *** *p* < 0.001. **m, n** Analysis of whole cell lysates (WCL) and extracellular vesicle (EV) fractions from control (NC) and *ATG9A*-depleted (si*ATG9A*#1) HEK-293T cells after 24-h DMEM starvation. **m** Immunoblotting for the indicated proteins. **n** Quantification of HNRNPK levels in EVs. Data are presented as mean ± SEM and analyzed using Student's t-test (*n* = 3 biologically independent experiments). ns not significant. **o, p** Control (NC) or *ATG9A*-depleted (si*ATG9A*#1) HeLa cells expressing galectin-9-Myc were treated with DMSO or 3-MA (1 mM). Galectin-9 secretion was analyzed by immunoblotting following 24-h DMEM starvation (**o**), and the relative levels were quantified (**p**). Asterisk indicates non-specific bands. Data are presented as mean ± SEM (*n* = 3 biologically independent experiments) and were analyzed by two-way ANOVA with Bonferroni multiple comparison test. *p < 0.05; **p < 0.01; ***p < 0.001.

including those of galectin-4, galectin-8, and annexin A6 (ANXA6), while showing no effect on IL-1β, galectin-3, or FGF2. These findings underscore the important role of ATG9A vesicles as selective carriers in unconventional protein secretion, with broad implications for understanding cellular processes and disease mechanisms.

## Results

### ATG9A is required for the unconventional secretion of galectin-9

Various stimuli, including serum deprivation[24], Poly(I:C)[25], and PMA[26], are commonly used to induce unconventional secretion of galectins. To determine whether ATG9A is critical for galectin-9 secretion, we utilized serum deprivation to induce unconventional secretion in HeLa cells and measured LDH release to assess cell death rate between groups. Knockdown or knockout of *ATG9A* reduced galectin-9 levels in the culture medium (Fig. 1a, b, and Supplementary Fig. 1a–d), suggesting a major role of ATG9A in mediating galectin-9 secretion.

To determine whether the impact of ATG9A on galetin-9 secretion is linked to its autophagic role, we examined the involvement of core autophagy-related (ATG) proteins that regulate different steps in the classical autophagy pathway. FIP200 and ULK1, which are essential for initiating classical autophagy[27], also contribute to an autophagy-independent lysosomal targeting pathway involving ATG9A[22]. Similarly, ATG2A and ATG2B, which play key roles in phagophore elongation through lipid transfer in conjunction with ATG9A[17,28], were investigated. Knockdown of *ULK1*, knockout of *FIP200*, or simultaneous depletion of *ATG2A* and *ATG2B* did not reduce galectin-9 secretion (Fig. 1c, d, and Supplementary Fig. 1e–j) but effectively suppressed autophagic flux (Supplementary Fig. 2), confirming the knockdown efficiency. These findings suggest that galectin-9 secretion operates independently of both the canonical autophagy pathway and the autophagy-independent lysosomal targeting pathway associated with ATG9A.

In contrast, the depletion of *LC3*, *ATG5*, or *ATG7*—key components of the LC3-conjugation machinery—significantly impaired galectin-9 secretion (Fig. 1e–h, and Supplementary Fig. 1k, l), demonstrating the critical role of LC3 lipidation in this process. LC3 lipidation occurs in various cellular pathways, including double membranes during canonical and secretory autophagy, and single membranes in conjugation of ATG8 to single membranes (CASM) pathways[29]. The LC3-dependent extracellular vesicle loading and secretion (LDELS) pathway shares multiple features with CASM, such as reliance on the LC3-conjugation system while being independent of FIP200 and the class III phosphoinositide 3-kinase (PI3K) complex[29–31]. Given our data and previous reports ruled out the involvement of canonical autophagy and CASM,

which functions independently of ATG9A[32–35], we next investigated whether ATG9A contributes to secretory autophagy or LDELS pathways. To test this, we examined the secretion of IL-1β, a prototypical secretory autophagy cargo[36], and HNRNPK, a representative LDELS cargo[30]. Colocalization analysis revealed that *ATG9A* knockout did not reduce IL-1β colocalization with LC3 (Fig. 1i, j), while *Atg5* knockout in mouse embryonic fibroblast (MEF) cells led to a significant decrease in IL-1β-positive secretory autophagosomes (Fig. 1k, l). Additionally, *ATG9A* knockdown had no effect on IL-1β secretion (Supplementary Fig. 3a, b), consistent with the prior findings that *ATG9A* knockout does not affect IL-1β secretion in response to starvation or lysosome damage in MEF cells[37]. Similarly, *ATG9A* depletion did not alter the secretion of the LDELS cargo HNRNPK (Fig. 1m, n), aligning with previous studies that LDELS depends on LC3-conjugation but not on other classical autophagosome formation components such as ATG14 and FIP200[30,31]. While our results establish that galectin-9 secretion requires LC3 lipidation, they also demonstrate that ATG9A's role in unconventional secretion is distinct from secretory autophagy or LDELS pathways. These findings suggest a distinctive and independent mechanism involving ATG9A in galectin-9 secretion.

Treatment with 3-methyladenine (3-MA), an inhibitor of the PI3K complex essential for autophagy but not required for LDELS or CASM[30–32,34], significantly reduced galectin-9 secretion (Fig. 1o, p), suggesting an important role of secretory autophagy in this process. Interestingly, both 3-MA treatment and *ATG9A* knockdown independently decreased galectin-9 secretion by ~50% (Fig. 1o, p). The additive reduction observed when 3-MA treatment was combined with *ATG9A* depletion suggests that secretory autophagy and the ATG9A-dependent secretion pathway function through distinct, non-redundant mechanisms.

### ATG9A vesicles serve as carriers for galectin-9 transport

Having established that ATG9A regulates galectin-9 secretion independently of canonical autophagy, secretory autophagy, LDELS, and CASM pathways, we investigated whether ATG9A vesicles, which are abundant in the cytoplasm and dynamically traffic between organelles, could serve as carriers for galectin-9 transport to the plasma membrane.

To determine whether intracellular galectin-9 is enclosed within membrane-bound structures in an ATG9A-dependent manner, we conducted a proteinase K protection assay[38,39]. The post-nuclear supernatant (PNS) was prepared from control and *ATG9A* knockout HeLa cells subjected to DMEM starvation to induce unconventional secretion. The 100k high-speed pellet (HSP) was isolated, resuspended, and divided into three equal fractions: untreated, proteinase

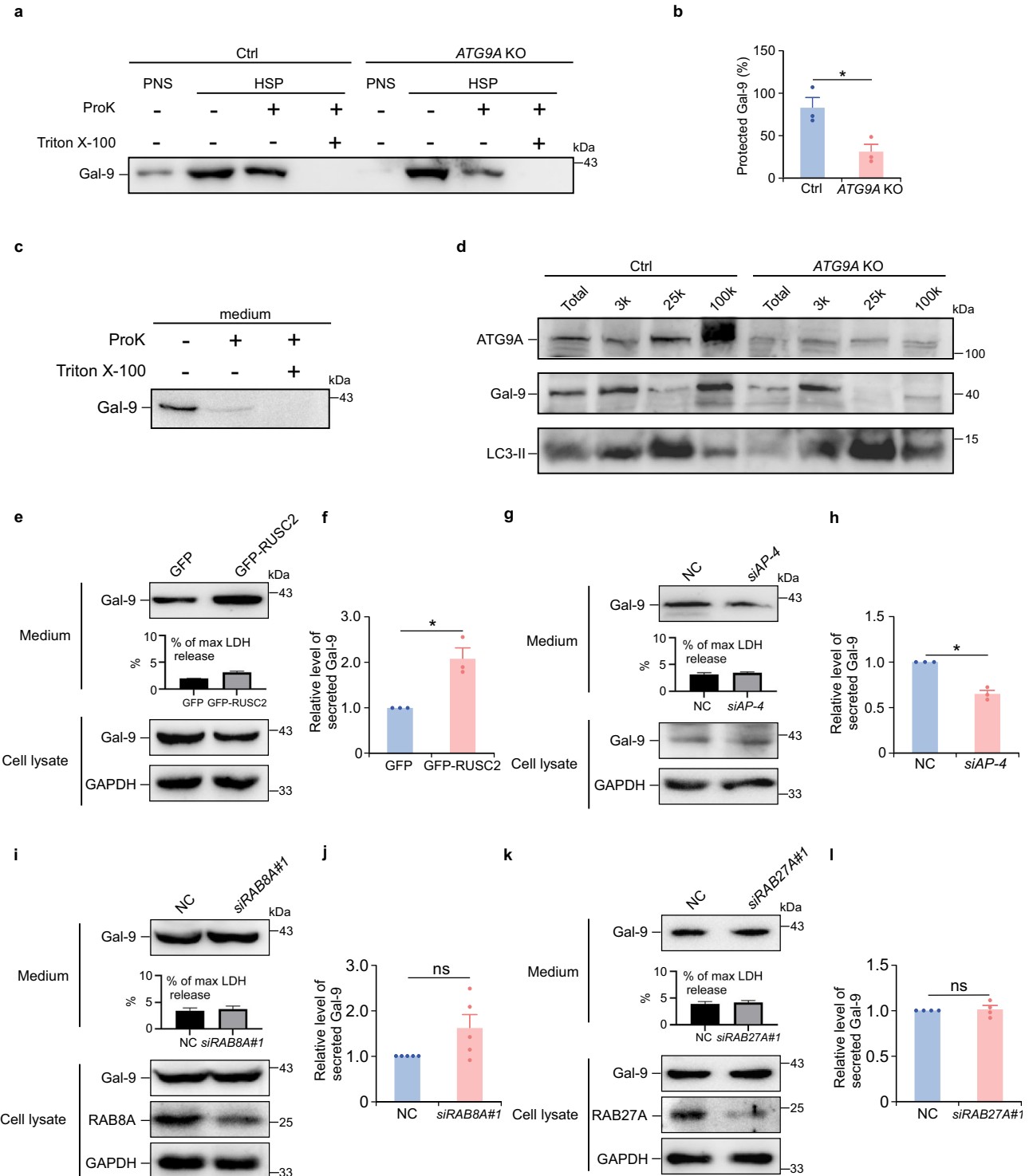

K-treated, and proteinase K plus detergent-treated. The assay revealed that intracellular galectin-9 was protected within membranous structures (Fig. 2a). In *ATG9A* knockout cells, the sensitivity of galectin-9 to proteinase K increased, and the amount of protected galectin-9 decreased (Fig. 2a, b), indicating that galectin-9 resides within ATG9A-dependent membranous structures. In contrast, minimal membrane protection was observed for galectin-9 in culture medium or the 100k supernatant (HSS) (Fig. 2c, and Supplementary Fig. 3c, d).

Having established that galectin-9 is protected within ATG9A-dependent membranous structures, we further investigated its association with ATG9A vesicle, using differential centrifugation analysis of

lysates from control and *ATG9A* knockout HeLa cells to separate distinct membranous components. Galectin-9, along with ATG9A, was enriched in the 100k sediments, while the prototypical secretory autophagy cargo IL-1β and autophagosome marker LC3 were predominantly found in the 25k sediments (Fig. 2d and Supplementary Fig. 3e). In *ATG9A* knockout cells, galectin-9 was scarcely detectable in the 100k sediments (Fig. 2d). As a control, IL-1β abundance in 25k sediments, corresponding to secretory autophagosomes remained unaffected by *ATG9A* knockout (Supplementary Fig. 3e). These findings underscore the crucial role of ATG9A vesicles in the transport and secretion of galectin-9.

**Fig. 2 | ATG9A vesicles serve as carriers for galectin-9 secretion. a, b** Proteinase K protection assay for galectin-9 in control (Ctrl) and *ATG9A* KO HeLa cells following 24-h DMEM starvation. **a** Immunoblot analysis of post-nuclear supernatant (PNS) and high-speed pellet (HSP) fractions incubated with or without proteinase K (Pro K) and Triton X-100. Quantification (**b**) shows the extent of membrane protection. Data are presented as mean ± SEM and analyzed using Student's t-test (*n* = 3 biologically independent experiments). *$p < 0.05$. **c** Proteinase K protection analysis of galectin-9 in conditioned medium harvested from HeLa cells after 24-h DMEM starvation. The conditioned medium was concentrated and treated with or without (Pro K) and Triton X-100. Representative data from 3 independent experiments are shown. **d** Immunoblot analysis of ATG9A, galectin-9, and LC3 in membrane fractions from control (Ctrl) and *ATG9A* KO HeLa cells isolated by differential centrifugation. Representative data from 3 independent experiments are shown. Galectin-9 secretion in GFP or GFP-RUSC2-expressing HeLa cells after 24-h DMEM starvation was analyzed by immunoblotting (**e**) and quantified (**f**). Data are presented as mean ± SEM and analyzed using Student's t-test (*n* = 3 biologically independent experiments). *$p < 0.05$. Galectin-9 secretion in control (NC) or *AP-4*-depleted (si*AP-4*) HeLa cells after 24-h DMEM starvation analyzed by immunoblotting (**g**) and quantified (**h**). Data are presented as mean ± SEM and analyzed using Student's t-test (*n* = 3 biologically independent experiments). *$p < 0.05$. Galectin-9 secretion in control (NC) or *RAB8A*-depleted (si*RAB8A*#1) HeLa cells following 24-h DMEM starvation was analyzed by immunoblotting (**i**) and quantified (**j**). Data are presented as mean ± SEM and analyzed using Student's t-test (*n* = 5 biologically independent experiments). ns not significant. Galectin-9 secretion in control (NC) or *RAB27A*-depleted (si*RAB27A*#1) HeLa cells following 24-h DMEM starvation was analyzed by immunoblotting (**k**) and quantified (**l**). Data are presented as mean ± SEM and analyzed using Student's t-test (*n* = 4 biologically independent experiments). ns not significant.

To explore the functional relationship between ATG9A vesicle trafficking and galectin-9 secretion, we manipulated key components of the ATG9A transport machinery. The export of ATG9A vesicles from the trans-Golgi network (TGN) to the cellular periphery is mediated by adapter protein 4 (AP-4) and its accessory protein RUSC2[40]. Supporting with the hypothesis that ATG9A vesicles transport galectin-9 for secretion, overexpression of *RUSC2*, which promotes the transport of ATG9A vesicles from the TGN to the plasma membrane[40], significantly enhanced galectin-9 secretion (Fig. 2e, f). Conversely, *AP-4* knockdown reduced galectin-9 secretion (Fig. 2g, h).

We also investigated the role of RAB27A and RAB8A, key regulators of multivesicular body (MVB), amphisome, and secretory lysosome transport to the plasma membrane[41–43]. Knockdown of *RAB8A* and *RAB27A* did not affect galectin-9 secretion (Fig. 2i–l, and Supplementary Fig. 4a–d), although the secretion of well-characterized unconventional secretory cargoes, such as HMGB1 and annexin A2 (ANXA2), was significantly reduced (Supplementary Fig. 4e–g). These findings suggest that galectin-9 secretion is independent of exosome- or secretory lysosome-mediated pathways. Together, these findings suggest that galectin-9 secretion is mediated by ATG9A vesicles, which encapsulate and transport galectin-9 for secretion.

## ATG9A N-terminus, galectin-9 CRDs and TMED10 orchestrate galectin-9 secretion

Immunofluorescence experiments revealed colocalization between galectin-9 and ATG9A under normal conditions, with a significant enhancement observed upon exposure to stimuli that induce unconventional galectin secretion (Fig. 3a, b). In contrast, colocalization between galectin-9 and LAMP1 decreased under the same condition (Supplementary Fig. 5). Protein co-immunoprecipitation and GST pull-down assays confirmed a direct interaction between ATG9A and galectin-9 (Fig. 3c, e), with the interaction significantly increasing in response to galectin secretion stimuli, such as serum deprivation or PMA treatment (Fig. 3c, d, and Supplementary Fig. 6).

To investigate the functional role of ATG9A domains in galectin-9 secretion, we overexpressed the full-length ATG9A, as well as its N-terminus and C-terminus truncations in *ATG9A* knockout cells. Deletion of the N-terminus of ATG9A, but not the C-terminus, significantly reduced galectin-9 secretion (Fig. 3f, g). Co-immunoprecipitation experiments with truncated ATG9A constructs revealed that removal of the N-terminus impaired the binding of galectin-9 to ATG9A (Fig. 3h, i). Considering the cytosolic orientation of ATG9A's N-terminus[17], these results suggest that the N-terminus of ATG9A plays a crucial role in facilitating galectin-9 binding and its entry into ATG9A vesicles, potentially via channel proteins.

Since galectin-9 is a tandem-repeat galectin containing two carbohydrate recognition domains (CRDs), we next assessed the contribution of each CRD to ATG9A-mediated secretion. Using Myc-tagged constructs expressing full-length galectin-9 or variants lacking either the N-terminal or C-terminal CRD, we analyzed galectin-9 secretion in control and *ATG9A*-depleted cells. While intracellular expression levels varied among the constructs, secretion pattern revealed a critical insight: deletion of either the N-terminal or C-terminal CRD abolished the ATG9A dependency of galectin-9 secretion (Fig. 3j–o). These results indicate that both CRDs are essential for ATG9A-mediated galectin-9 secretion.

Transmembrane emp24 domain-containing protein 10 (TMED10), previously identified as a channel for unconventional secretion cargo entering the ER-Golgi intermediate compartment (ERGIC)[39], emerged as a potential facilitator of galectin-9 secretion. We found that upon poly(I:C)-induced secretion, TMED10 colocalized with ATG9A and galectin-9 (Fig. 4a–c), suggesting its association with ATG9A vesicles. Furthermore, in vivo interaction between TMED10 and galectin-9 was confirmed (Fig. 4d). *TMED10* knockdown significantly reduced galectin-9 secretion (Fig. 4e, f, and Supplementary Fig. 7a, b), as well as IL-1β secretion and its colocalization with LC3, validating the efficiency of the knockdown (Supplementary Fig. 7c–f). Importantly, *TMED10* knockdown also reduced the amount of membrane-protected galectin-9 and decreased colocalization between galectin-9 and ATG9A (Fig. 4g–j). These findings indicate that TMED10 facilitates the entry of galectin-9 into ATG9A vesicles, enabling its subsequent transport and secretion.

## The STX13-SNAP23-VAMP3 SNARE complex is involved in the fusion of ATG9A vesicles with the plasma membrane

Previous studies have demonstrated that ATG9A vesicles can translocate to the site of plasma membrane damage and fuse with the plasma membrane through a SNARE-dependent manner[19]. However, the specific SNARE complex mediating this fusion process has not been clearly identified. To investigate this, we used a well-established propidium iodide (PI) uptake assay to screen SNAREs potentially involved in the fusion of ATG9A vesicles with the plasma membrane[19,44]. Knockdown of *STX13*, *SNAP23*, or *VAMP3* led a significantly increased plasma membrane damage compared to knockdown of other SNAREs (Fig. 5a–f). Cell surface biotinylation experiments further revealed that plasma membrane damage caused a notable translocation of ATG9A to cell surface. However, this translocation was significantly diminished upon knockdown of *STX13*, *SNAP23*, or *VAMP3* (Fig. 5g, h). These results align with previous findings implicating the STX13-SNAP23-VAMP3 SNARE complex in secretion[45] and lamellipodium extension[46], suggesting that this SNARE complex facilitates the fusion of ATG9A vesicles with the plasma membrane.

As carriers of galectin-9, ATG9A vesicles must also fuse with the plasma membrane to release cargo into the extracellular space. To explore the role of the STX13-SNAP23-VAMP3 complex in this process, we analyzed galectin-9 secretion following knockdown of *STX13, SNAP23, or VAMP3*. Knockdown of *STX13*, *SNAP23*, or *VAMP3* significantly reduced galectin-9 secretion in HeLa cells (Fig. 6a–f). Further validation was performed using mutant forms of SNAP23 (lacking nine

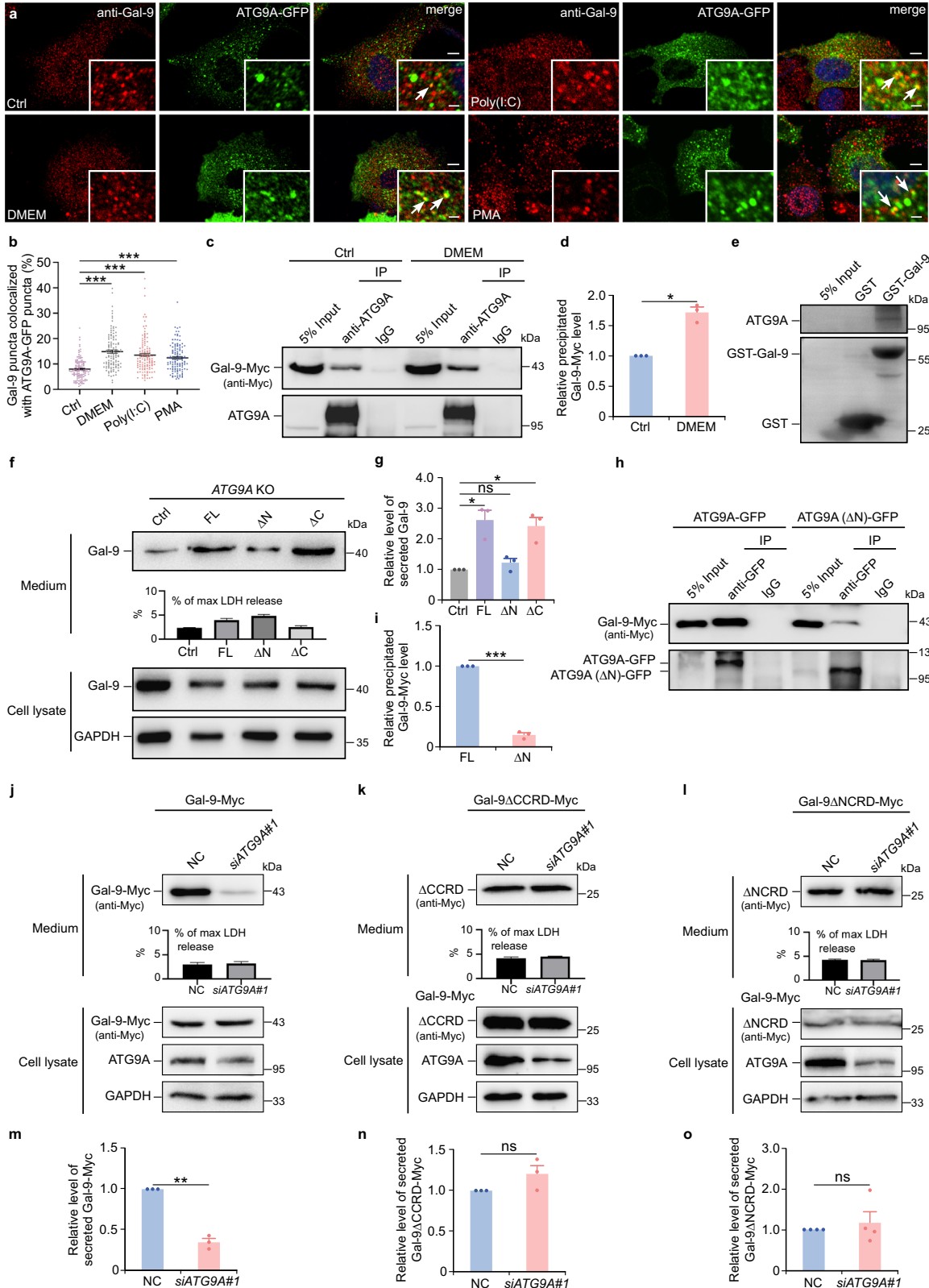

C-terminus amino acids), STX13, and VAMP3 (both lacking trans-membrane domains), which blocked membrane fusion. These mutants effectively inhibited galectin-9 secretion in HEK-293T cells (Supplementary Fig. 8a–f), confirming that membrane fusion mediated by the STX13-SNAP23-VAMP3 complex is critical for galectin-9 release. In contrast, knockdown of *SEC22B* and *YKT6*, the SNAREs known to regulate the fusion of secretory autophagosomes and MVBs with plasma membranes, did not affect galectin-9 secretion (Fig. 6g, h, and Supplementary Fig. 8g–j). Additionally, knockdown of *STX13* or *SNAP23* led to increased intracellular accumulation of galectin-9 and enhanced colocalization of galectin-9 with ATG9A, while *SEC22B* knockdown had no such effect (Fig. 6i–k). Upon induction of galectin-9 secretion, colocalization between ATG9A and VAMP3 increased (Fig. 6l, m), and *SNAP23* knockdown caused VAMP3-positive ATG9A vesicles to

**Fig. 3 | N-terminus of ATG9A and CRDs of galectin-9 are required for galectin-9 secretion. a, b** Immunostaining of galectin-9 in HeLa cells expressing ATG9A-GFP under normal conditions (Ctrl) or after treatment with DMEM, Poly(I:C) (20 μg/ml), or PMA (60 ng/ml) for 24 h. **a** The arrows point to galectin-9 puncta colocalized with ATG9A puncta. Scale bars, 5 μm, inserts, 1 μm. **b** The percentage of galectin-9 puncta colocalized with ATG9A-GFP puncta was quantified. Data represent mean ± SEM and analyzed using Wilcoxon rank sum test ($n = 120$ cells in each group examined over 3 independent experiments). *** $p < 0.001$. **c, d** ATG9A was immunoprecipitated using an anti-ATG9A antibody from HeLa cells expressing galectin-9-Myc under normal conditions or following 24-h DMEM starvation. **c** The resultant precipitates were subjected to immunoblotting with anti-Myc and anti-ATG9A antibodies. **d** The levels of galectin-9-Myc (normalized to ATG9A levels) were quantified. Data represent mean ± SEM and analyzed using Student's t-test ($n = 3$ biologically independent experiments). *$p < 0.05$. **e** Purified GST or GST-Gal-9 was incubated with HeLa cell lysates, followed by immunoblotting with anti-ATG9A and anti-GST antibodies. Representative data from 3 independent experiments are shown. **f, g** Galectin-9 secretion in *ATG9A* KO HeLa cells expressing GFP (Ctrl), GFP-tagged ATG9A (FL), or indicated variants (ΔN, aa 59-839; ΔC, aa 1-505). Cells were starved for 24 h with DMEM. **f** Immunoblot analysis. **g** Quantification of secreted galectin-9 levels. Data are presented as mean ± SEM and analyzed using Student's t-test ($n = 3$ biologically independent experiments). *$p < 0.05$; ns not significant. **h, i** Galectin-9-Myc was co-expressed with ATG9A (ΔN)-GFP or ATG9A-GFP in HeLa cells. The cell lysates were immunoprecipitated with an anti-GFP antibody. The resulting precipitate was immunoblotted with anti-Myc and anti-GFP antibodies (**h**). **i** Quantification of the relative levels of galectin-9-Myc (normalized by ATG9A-GFP or ATG9A (ΔN)-GFP levels) was shown as mean ± SEM and analyzed using Student's t-test ($n = 3$ biologically independent experiments). ***$p < 0.001$. **j–o** Analysis of protein secretion in control (NC) and *ATG9A*-depleted (si*ATG9A*#1) HEK-293T cells expressing Myc-tagged galectin-9 (FL) or variants (ΔNCRD or ΔCCRD) after 24-h DMEM starvation. **j–l** Immunoblot analysis. **m–o** Quantification. Data are presented as mean ± SEM and analyzed using Student's t-test ($n = 3$ biologically independent experiments for **m**, **n**, $n = 4$ biologically independent experiments for **o**). ** $p < 0.01$; ns not significant.

accumulate in the cytoplasm (Fig. 6n, o). Together, these findings demonstrate that the STX13-SNAP23-VAMP3 SNARE complex is essential for mediating the fusion of ATG9A vesicles with the plasma membrane. This complex plays a key role in regulating both plasma membrane repair and the unconventional secretion of galectin-9.

## ATG9A vesicles mediate the selective secretion of multiple unconventional cargo proteins

The findings above highlight ATG9A vesicles as a unique carrier for the unconventional secretory cargo galectin-9. This prompted us to investigate whether ATG9A vesicles might also mediate the secretion of other unconventional cargo proteins. The secretion of galectin-4 and galectin-8, tandem-repeat galectins containing two CRDs similar to galectin-9, was shown to be ATG9A-dependent (Fig. 7a–d). In contrast, the secretion of galectin-3, a chimera-type galectin with a single CRD, was unaffected by *ATG9A* knockdown (Fig. 7e, f). These findings were consistent across HEK-293T cells (Supplementary Fig. 9a–h). Proteinase K sensitivity assays revealed that intracellular galectin-4 and galectin-8, but not galectin-3, were protected by membrane structures in an ATG9A-dependent manner (Fig. 7g–l). This aligns with the observation that removing either CRD from galectin-9, which converts it into a form resembling single-CRD galectins, abolishes its ATG9A-dependent secretion (Fig. 3j–o). This finding suggests that the tandem-repeat structure of galectins is crucial for their secretion via ATG9A vesicles. Immunofluorescence analysis showed that galectin-8 displayed a punctate distribution pattern, with ~20% of these puncta colocalizing with ATG9A but not with galectin-9 (Fig. 7m, n). Only 6% of the puncta showed triple colocalization of galectin-8, galectin-9, and ATG9A (Fig. 7m, n), suggesting that ATG9A may interact independently with galectin-8 and galectin-9, despite some shared trafficking routes. In contrast, galectin-4 exhibited a diffuse cytoplasmic distribution (Supplementary Fig. 10) like what was described in previous literatures[47,48], with no apparent punctate structures, making it challenging to determine whether galectin-4 is co-transported with galectin-9 by ATG9A vesicles.

Building on this, we explored whether ATG9A vesicles mediate the secretion of other unconventional cargo proteins. Indeed, annexin A6 secretion was shown to depend on ATG9A, as demonstrated by a significant reduction in annexin A6 secretion in *ATG9A* knockout cells (Fig. 8a, b). In contrast, knockdown of *RAB27A*, *RAB8A*, or *YKT6* had no significant effect on annexin A6 secretion (Fig. 8c–h). Similar to galectin-9 secretion, *RUSC2* overexpression markedly enhanced annexin A6 secretion (Fig. 8i, j). We also investigated the role of STX13-SNAP23-VAMP3 SNARE complex in annexin A6 secretion. Knockout of *STX13* reduced annexin A6 secretion (Fig. 8k, l). Moreover, SNAP23 and VAMP3 mutants, lacking the C-terminal nine amino acids or the transmembrane domain, respectively, blocked annexin A6 secretion

effectively (Fig. 8m–p). These results indicate that membrane fusion mediated by the STX13-SNAP23-VAMP3 SNARE complex is crucial for annexin A6 secretion, further underscoring the shared mechanistic framework of ATG9A-dependent secretion. As a control, Fibroblast Growth Factor 2 (FGF2), a type I UPS cargo protein that undergoes direct plasma membrane translocation, was secreted independently of ATG9A (Supplementary Fig. 9i, j). This finding confirms the specificity of the ATG9A-dependent secretory pathway.

Collectively, our results demonstrate that the ATG9A vesicle-mediated secretory pathway transports a diverse range of cargo proteins, including galectin-8, galectin-4, and annexin A6, while selectively excluding others, such as galectin-3 and FGF2. This pathway exhibits both versatility and specificity in cargo selection. Moreover, the molecular machinery governing this process, including RUSC2 and the STX13-SNAP23-VAMP3 SNARE complex, is conserved across different cargo proteins, suggesting a common mechanistic basis for ATG9A-dependent unconventional protein secretion.

## ATG9A-mediated galectin-9 transport is conserved in monocytic cells

Previous studies have shown that galectin-9 interacts with VAMP3 and regulates the secretion of interleukin-6 (IL-6) and other cytokines in monocytic cells[49]. To determine whether this regulatory role of galectin-9 is connected to the ATG9A-mediated unconventional secretion pathway we identified, we first examined the effect of ATG9A on IL-6 secretion in THP-1 cells, a well-established human monocytic cell line that widely used to study monocyte, macrophage and dendritic cell biology. Interestingly, *ATG9A* knockdown had no impact on IL-6 secretion in either THP-1 or HEK-293T cells (Fig. 9a, b, and Supplementary Fig. 11a, b), suggesting that the ATG9A-mediated galectin-9 secretory pathway is distinct from the mechanism through which galectin-9 regulates IL-6 secretion. In THP-1 cells, galectin-9 is not only secreted but also remains bound to the extracellular side of the plasma membrane. Notably, *ATG9A* knockdown in THP-1 cells reduced both galectin-9 secretion and its attachment to the plasma membrane (Fig. 9c–f). Overall, our results reveal that the ATG9A-mediated transport and distribution of galectin-9 represents a fundamental cellular process. This mechanism operates independently of galectin-9's role in cytokine regulation and is potentially conserved across other cell types, such as monocytic cells.

## Discussion

Although ATG9A was initially identified as a core autophagy protein, its ubiquitous localization and the dynamic nature of ATG9A vesicles suggest functional diversity. This study introduces ATG9A vesicles as a critical type III unconventional secretion carrier, specifically facilitating the transport of galectin-9 and other cargo proteins. Mechanistically,

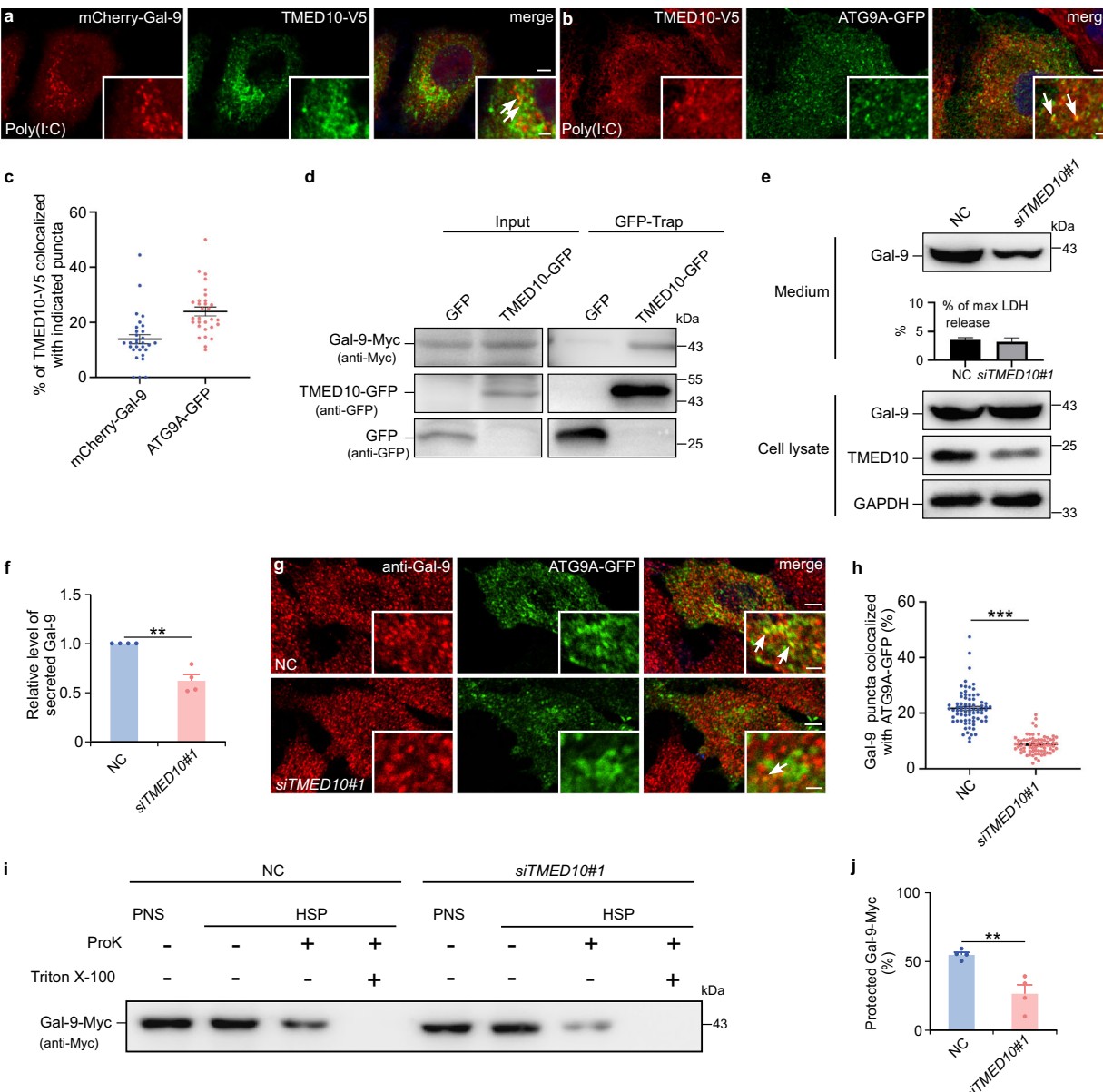

**Fig. 4 | TMED10 mediates galectin-9 secretion. a** Fluorescence images of HeLa cells expressing both mCherry-galectin-9 and TMED10-V5, immunostained with an anti-V5 antibody. Cells were treated with Poly(I:C) (20 μg/ml) for 24 h. Arrows indicate mCherry-galectin-9 puncta colocalized with TMED10-V5 puncta. Scale bars, 5 μm, inserts, 2 μm. **b** Fluorescence analysis of HeLa cells expressing TMED10-V5 and ATG9A-GFP, immunostained with an anti-V5 antibody. Cells were treated with Poly(I:C) (20 μg/ml) for 24 h. Arrows indicate ATG9A-GFP puncta colocalized with TMED10-V5 puncta. Scale bars, 5 μm; inserts, 2 μm. **c** Quantification of TMED10-V5 puncta colocalized with mCherry-galectin-9 or ATG9A-GFP puncta. Data are presented as mean ± SEM (*n* = 30 cells per group examined over 3 independent experiments). **d** Galectin-9-Myc was co-expressed with TMED10-GFP in HeLa cells. The lysates were precipitated using GFP-Nanoab-Agarose beads. The precipitate was immunoblotted with anti-Myc and anti-GFP antibodies. Representative data from 3 independent experiments are shown. Galectin-9 secretion in control (NC) or *TMED10*-depleted (si*TMED10*#1) HeLa cells following 24-h DMEM starvation analyzed by immunoblotting (**e**) and quantified (**f**). Data are presented as mean ± SEM and analyzed using Student's t-test (*n* = 4 biologically independent experiments). ** *p* < 0.01. **g, h** Immunostaining of galectin-9 in control (NC) or *TMED10*-depleted (si*TMED10*#1) HeLa cells expressing ATG9A-GFP following 24-h DMEM starvation. **g** Arrows indicate galectin-9 puncta colocalized with ATG9A-GFP puncta. Scale bars, 5 μm, inserts, 1 μm. **h** The percentage of galectin-9 puncta colocalized with ATG9A-GFP puncta was quantified. Data are presented as mean ± SEM and analyzed using Wilcoxon rank sum test (*n* = 75 cells in each group examined over 3 independent experiments). ****p* < 0.001. **i, j** Proteinase K protection assay for galectin-9-Myc in control (NC) and *TMED10*-depleted (si*TMED10*#1) HeLa cells following 24-h DMEM starvation. **i** Immunoblot analysis of post-nuclear supernatant (PNS) and high-speed pellet (HSP) fractions incubated with or without Pro K and Triton X-100. **j** Quantification of galectin-9 protection levels from (**i**). Data are presented as mean ± SEM and analyzed using Student's t-test (*n* = 4 biologically independent experiments). ** *p* < 0.01.

ATG9A interacts with galectin-9 via its N-terminus, with both CRDs of galectin-9 being essential for its ATG9A-mediated secretion. Additionally, TMED10 facilitates cargo translocation into ATG9A vesicles, RUSC2 promotes ATG9A vesicle trafficking to the plasma membrane, and the STX13-SNAP23-VAMP3 SNARE complex mediates the fusion of ATG9A vesicles with the plasma membrane (Fig. 9g). Importantly, this

ATG9A vesicle-mediated secretion mechanism is potentially conserved across other cell types, such as monocytic cells, underscoring its critical role in cellular protein trafficking.

While ATG9A vesicles are best known as seed membrane structures in the autophagy pathway[50], whether they transport cargo and, if so, what cargo they carry has remained unclear. In this study, we

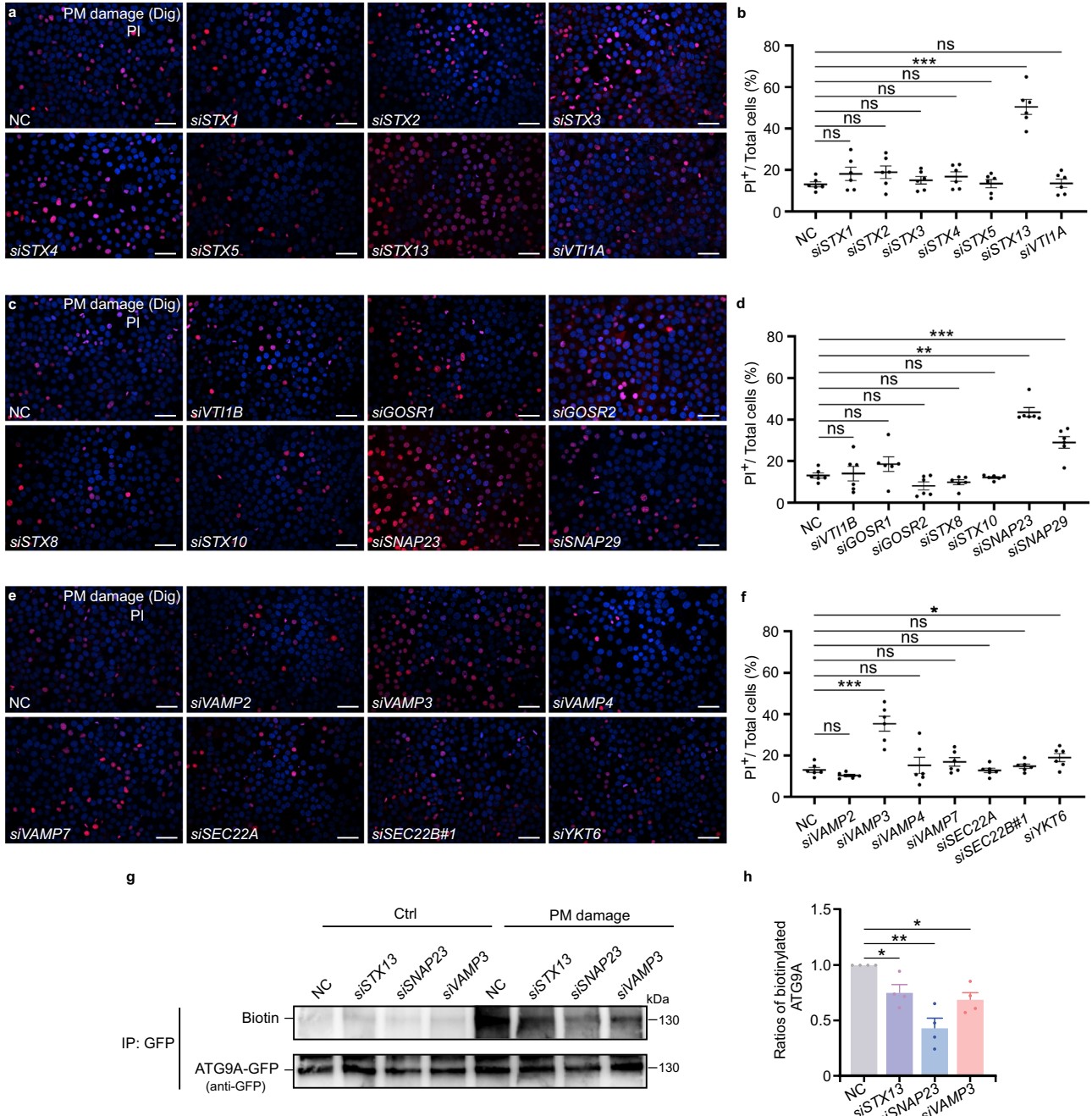

**Fig. 5 | The STX13-SNAP23-VAMP3 SNARE complex is involved in the fusion of ATG9A vesicles with PM in protection against PM damage. a** Confocal micro-scopy analysis of plasma membrane permeabilization after knockdown of *STX1, STX2, STX3, STX4, STX5, STX13,* or *VTI1A* in HeLa cells. Cells were stained with pro-pidium iodide (PI) following plasma membrane damage induced by digitonin (Dig). Scale bars, 50 μm. Representative data from 6 independent experiments are shown. **b** Quantification of plasma membrane permeabilization induced by digitonin after knockdown of the indicated genes. The percentage of PI⁺ (nuclei) cells in (**a**) is shown as mean ± SEM and analyzed using Student's t-test and Welch's t-test (*n* = 6 biologically independent experiments). ns not significant; \*\*\**p* < 0.001. **c** Confocal microscopy analysis of plasma membrane permeabilization after knockdown of *VTI1B, GOSR1, GOSR2, STX8, STX10, SNAP23,* or *SNAP29.* Cells were stained with propidium iodide (PI) following plasma membrane damage induced by digitonin (Dig). Scale bars, 50 μm. Representative data from 6 independent experiments are shown. **d** Quantification of plasma membrane permeabilization induced by digi-tonin after knockdown of the indicated genes. The percentage of PI⁺ (nuclei) cells in (**c**) is shown as mean ± SEM and analyzed using Student's t-test, Welch's t-test, and Wilcoxon rank sum test (*n* = 6 biologically independent experiments). ns not

significant; \*\**p* < 0.01; \*\*\**p* < 0.001. **e** Confocal microscopy analysis of plasma membrane permeabilization after knockdown of *VAMP2, VAMP3, VAMP4, VAMP7, SEC22A, SEC22B*#1, or *YKT6.* Cells were stained with propidium iodide (PI) following plasma membrane damage induced by digitonin (Dig). Scale bars, 50 μm. Repre-sentative data from 6 independent experiments are shown. **f** Quantification of plasma membrane permeabilization induced by digitonin after knockdown of the indicated genes. The percentage of PI⁺ (nuclei) cells in (**e**) is shown as mean ± SEM and analyzed using Student's t-test (*n* = 6 biologically independent experiments). ns not significant; \**p* < 0.05; \*\*\**p* < 0.001. **g** Cell surface biotinylation analysis of ATG9A in control (NC), *STX13*-depleted (si*STX13*), *SNAP23*-depleted (si*SNAP23*), or *VAMP3*-depleted (si*VAMP3*) HeLa cells expressing ATG9A-GFP (under normal con-ditions or after plasma membrane damage induced by Dig). ATG9A-GFP was immunoprecipitated with an anti-GFP antibody and analyzed by Western blot using Streptavidin-HRP antibody. Representative data from 4 independent experiments are shown. **h** Quantification of the levels of biotinylated ATG9A (normalized to ATG9A-GFP levels) is shown as means ± SEM and analyzed using Student's t-test (*n* = 4 biologically independent experiments). \**p* < 0.05; \*\**p* < 0.01.

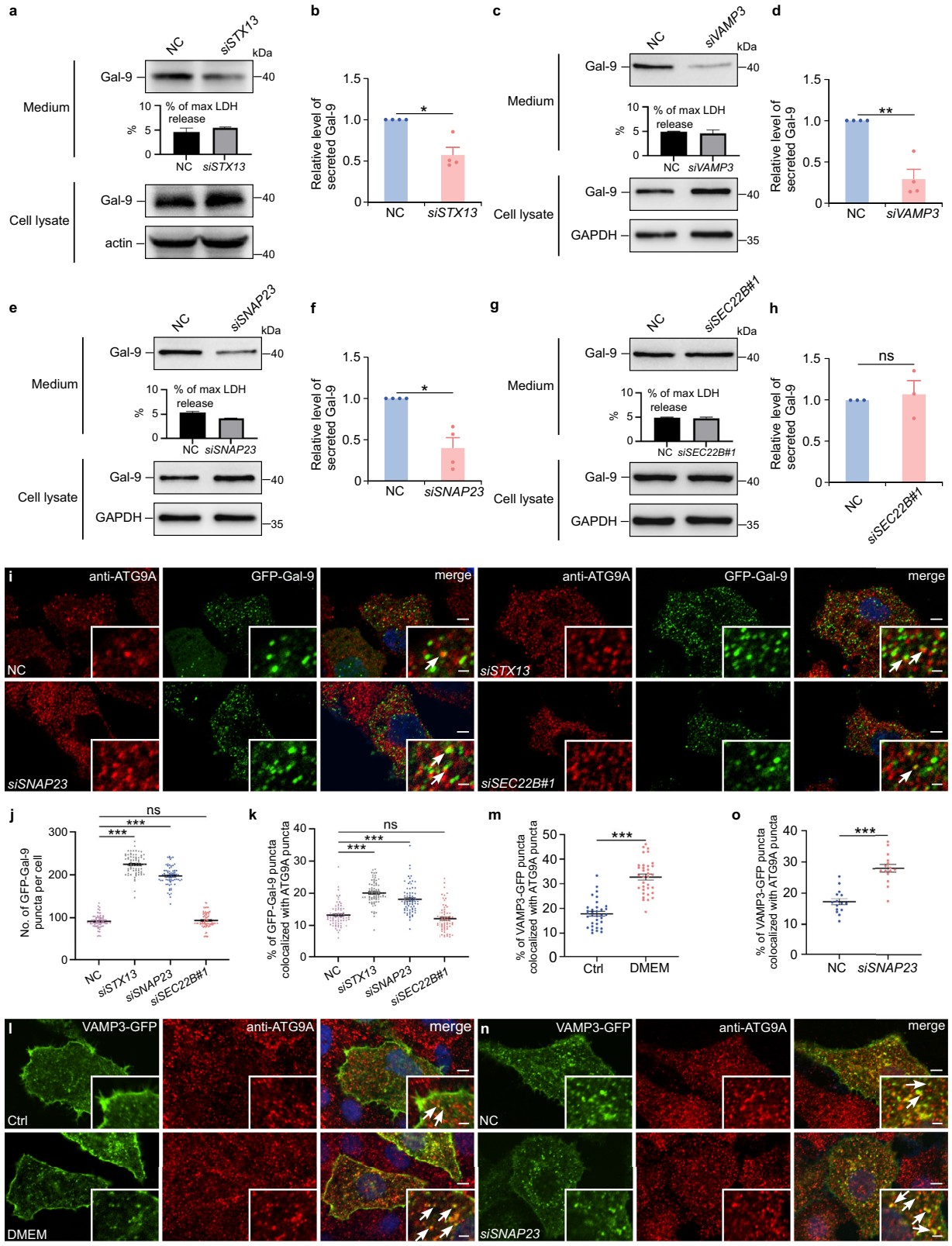

uncover a critical vesicular transport function of ATG9A, demonstrating its role in the unconventional secretion of a diverse array of proteins, including galectin-9, galectin-4, galectin-8, and annexin A6, while showing no effect on IL-1β, galectin-3, or FGF2. Notably, all ATG9A-dependent galectins (galectin-4, -8, and -9) belong to the tandem-repeat type, whereas galectin-3, which has only one CRD, is not regulated by ATG9A. This aligns with the observation that deleting either CRD from galectin-9—thereby converting it into a form resembling single-CRD galectins—abolishes its ATG9A-dependent secretion. This finding suggests that the tandem-repeat structure is crucial for this secretion. Together, these findings indicate a finely tuned cargo-sorting mechanism, with evidence of direct interaction between the cytoplasmic N-terminal domain of ATG9A and galectin-9. However, further investigation is needed to identify additional factors involved

**Fig. 6 | The STX13-SNAP23-VAMP3 SNARE complex is involved in galectin-9 secretion facilitated by ATG9A vesicles. a, c, e, g** Galectin-9 secretion in control (NC) or *STX13*-depleted (si*STX13*), *VAMP3*-depleted (si*VAMP3*), *SNAP23*-depleted (si*SNAP23*), *SEC22B*-depleted (si*SEC22B*) HeLa cells starved with DMEM for 24 h. **b, d, f, h** Quantification of secreted galectin-9 levels. Results represent mean ± SEM and analyzed using Student's t-test ($n = 4$ biologically independent experiments in **b, d, f**, $n = 3$ biologically independent experiments in **h**). ns not significant; *$p < 0.05$; **$p < 0.01$. **i–k** Control (NC), *STX13*-depleted (si*STX13*), *SNAP23*-depleted (si*SNAP23*), or *SEC22B*-depleted (si*SEC22B*) HeLa cells were transfected with GFP-galectin-9 plasmid and immunostained for ATG9A. Arrows indicate GFP-galectin-9 puncta colocalized with ATG9A puncta. Scale bars, 5 μm, inserts, 1 μm. Quantification of GFP-galectin-9 puncta (**j**) and the percentage of GFP-galectin-9 puncta colocalized with ATG9A puncta (**k**) are depicted. Data represent mean ± SEM and analyzed using Welch's t-test and Wilcoxon rank sum test ($n = 72$ cells in each group

examined over 3 independent experiments). ns not significant; ***$p < 0.001$. **l, m** HeLa cells expressing VAMP3-GFP were subjected to DMEM starvation for 24 h and then immunostained for endogenous ATG9A (**l**). Arrows indicate VAMP3-GFP puncta colocalized with ATG9A puncta. **m** Colocalization was quantified, with data presented as mean ± SEM and analyzed using Welch's t-test ($n = 56$ cells in each group examined over 3 independent experiments). Scale bars, 5 μm; inserts, 2 μm. ***$p < 0.001$. **n, o** Immunostaining of ATG9A in control (NC) and *SNAP23*-depleted (si*SNAP23*) HeLa cells expressing VAMP3-GFP following 24-h DMEM starvation. **n** Representative images are shown. Arrows indicate VAMP3-GFP puncta colocalized with ATG9A puncta. Scale bars, 5 μm; inserts, 2 μm. **o** Quantification of the percentage of VAMP3-GFP puncta colocalized with ATG9A puncta. Data represent mean ± SEM and analyzed using Student's t-test ($n = 15$ cells in each group examined over 3 independent experiments). ***$p < 0.001$.

in cargo recruitment and sorting. Future studies should also clarify the distinctions and overlaps in vesicle formation, cargo loading, and targeted transport between ATG9A vesicles involved in autophagy and those implicated in unconventional secretion. A thorough investigation is essential to uncover the upstream regulatory factors governing ATG9A vesicles and their involvement in unconventional secretion.

Galectin-9, despite its critical roles in cellular adhesion, communication, differentiation, and survival[9], and its strong associations with human diseases[9,11], remains poorly understood in terms of its unconventional secretion mechanism. Our findings identify ATG9A vesicles as a key player in galectin-9 secretion. Interestingly, secretory autophagosomes also participate in galectin-9 secretion independently of ATG9A. Although galectin-9 has been detected on exosomes in Epstein-Barr virus-infected nasopharyngeal carcinoma cells[51] and its secretion is inhibited by exosome formation blockers[52], knockdown of exosome pathway regulators has minimal impact on galectin-9 secretion in HeLa cells (Fig. 2k, l, and Supplementary Figs. 4c, d and 8g, h). These observations suggest that the regulation of galectin-9 secretion varies across cell types and under different induction conditions. Further research is needed to elucidate how ATG9A vesicle, secretory autophagosome, and exosome coordinate to regulate galectin-9 secretion under distinct physiological and pathological contexts. It's tempting to speculate that such coordination enables cells to adapt their secretion mechanism to specific environment or stress conditions.

The vesicular transport function of ATG9A likely extends beyond unconventional secretion. For example, ATG9A has been implicated in plasma membrane repair[19], a process involving annexins, which are also cargo protein in ATG9A vesicles. This raises the intriguing possibility that both the ATG9A and its vesicular cargo synergize in membrane repair. Unconventional secretion is closely linked to cellular response to stress and pathogenic stimuli. Galectins and annexins, for instance, are upregulated during immune responses and infections, processes in which ATG9A has also been reported to play a regulatory role, such as modulating innate immune response to dsDNA[53]. These observations suggest that ATG9A vesicles may contribute to disease pathogenesis through unconventional secretion pathways. Understanding the interplay between ATG9A vesicles and their vesicular cargos under diverse physiological and pathological conditions not only deepens our knowledge of ATG9A-mediated vesicular transport but also sheds light on how cells fine-tune vesicle-mediated secretion to maintain homeostasis and respond to diseases.

## Methods
### Antibodies
Antibodies used in this study: rabbit anti-galectin-9 (abcam, ab69630, 1:1000), rabbit anti-galectin-9 (Thermo Fisher Scientific, PA5-115266, 1:1000), rabbit anti-galectin-9 (abcam, ab227046, 1:1000), goat anti-galectin-9 (AF2045, R&D systems, 1:500), mouse anti-GAPDH (proteintech, 60004-1-Ig, 1:10000), rabbit anti-ATG2A (MBL, PD041,

1:1000), rabbit anti-ATG2B (proteintech, 25155-1-AP, 1:1000), rabbit anti-ATG9A (MBL, PD042, 1:1000 for western blot and 1:200 for immunofluorescence staining), mouse anti-GST (proteintech, 66001-2-Ig, 1:2000), rabbit anti-Myc (proteintech, 16286-1-AP, 1:2000 for western blot and 1:200 for immunofluorescence staining), rabbit anti-GFP (proteintech, 50430-2-AP, 1:2000), mouse anti-annexin A6 (Santa Cruz Biotechnology, sc-271859, 1:1000), mouse anti-actin (proteintech, 66009-1-Ig, 1:2000), mouse anti-V5 (Thermo Fisher Scientific, R960-25, 1:500), rabbit anti-FLAG (proteintech, 20543-1-AP, 1:10000), rabbit anti-IL-1β (abcam, ab9722, 1:1000 for western blot and 1:200 for immunofluorescence staining), rabbit anti-LC3B (Cell Signaling Technology, 2775S, 1:1000), Mouse monoclonal anti-LC3 (MBL, M152-3, 1:200), mouse anti-TMED10 (Proteintech, 67876-1-lg, 1:1000), rabbit anti-RAB27A (Proteintech, 17817-1-AP, 1:1000), rabbit anti-RAB8A (Proteintech, 55296-1-AP, 1:1000), rabbit anti-SEC22B (Proteintech, 14776-1-AP, 1:1000), rabbit anti-ULK1(Sigma, A7481, 1:1000), mouse anti-HNRNPK (Proteintech, 67708-1-Ig, 1:1000), FITC AffiniPure Goat anti-Mouse IgG (H+L) (Jackson, 115-095-003, 1:400), Alexa Fluor® 647 AffiniPure™ Donkey Anti-Mouse IgG (H+L) (Jackson, 715-605-150, 1:400), Rhodamine AffiniPure Goat anti-Rabbit IgG (Jackson, 111-025-003, 1:400), Rhodamine AffiniPure Goat anti-Mouse IgG (Jackson, 115-025-003, 1:400), HRP-conjugated Affinipure Goat Anti-Rabbit IgG (H+L) (proteintech, SA00001-2, 1:5000), HRP-conjugated Affinipure Goat Anti-mouse IgG (H+L) (proteintech, SA00001-1, 1:5000), mouse anti-Biotin (Santa Cruz Biotechnology, sc-53179, 1:500).

### Cell culture
HeLa, HEK-293T, and MEF cells were cultured in Dulbecco's Modified Eagle Medium (DMEM) supplemented with 10% fetal bovine serum (FBS), penicillin, and streptomycin at 37 °C in a humidified atmosphere with 5% $CO_2$. THP-1 was cultured in RPMI Medium 1640 basic (RPMI-1640) supplemented with 10% fetal bovine serum (FBS), penicillin, streptomycin, and β-mercaptoethanol (0.05 mM) at 37 °C in a humidified atmosphere with 5% $CO_2$. For serum starvation experiments, HeLa, HEK-293T, or MEF cells were washed three times with 1× phosphate-buffered saline (PBS) and cultured in DMEM without FBS, and THP-1 cells were cultured in RPMI-1640. For treatment with 3-MA (Apexbio, A8353), cells were treated with 1 mM 3-MA for 6 h. For treatment with phorbol 12-myristate 13-acetate (PMA, ALADDIN, P167764), Polyinosinic-polycytidylic acid (Poly(I:C), Apexbio, B5551), cells were incubated in DMEM supplemented with 10% FBS and either PMA (60 ng/ml) or Poly(I:C) (20 μg/ml) for 24 h. *ATG9A* knockout and *FIP200* knockout HeLa cells, as well as *Atg5* knockout MEF cells, were generously provided by Dr Hong Zhang from the Institute of Biophysics, Chinese Academy of Sciences.

### Plasmids
Full-length cDNA of human galectin-9, galectin-4, galectin-8, ANXA2, and HMGB1 were PCR amplified and cloned into the pcDNA3.1 vector with a C-terminal 3×Myc tag, resulting in the plasmids galectin-9-Myc,

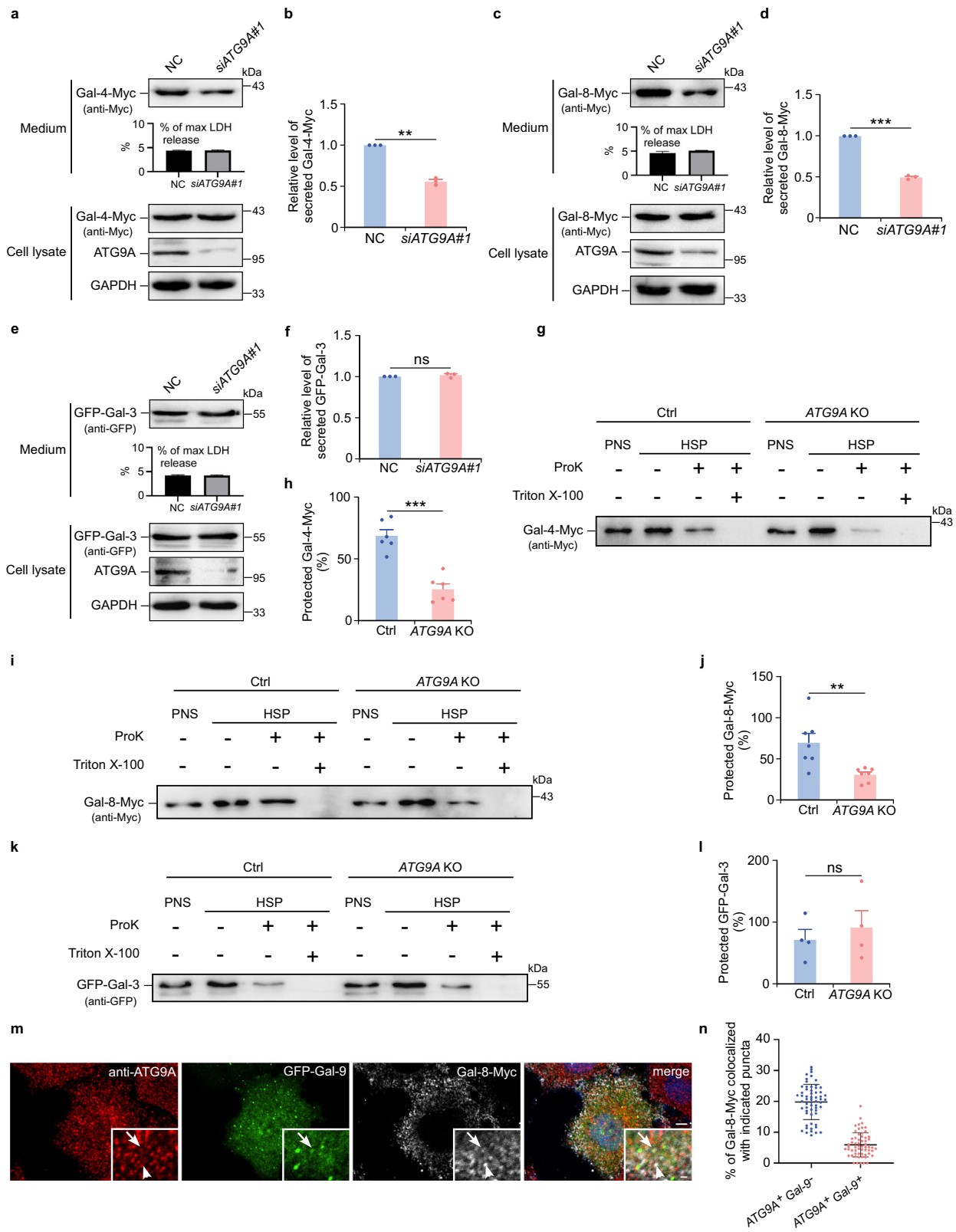

galectin-4-Myc, galectin-8-Myc, ANXA2-Myc, and HMGB1-Myc. The galectin-9 cDNA was subsequently subcloned into pEGFP-C1, mCherry-C1, and pGEX-5X-2 to generate the constructs GFP-galectin-9, mCherry-galectin-9, and GST-galectin-9, respectively. Truncated variants of galectin-9 and ATG9A were produced via site-directed mutagenesis. ATG9A-GFP and TMED10-GFP constructs were created by subcloning their cDNA into pcDNA3.1 with a C-terminal GFP tag. The

cDNA of human STX13, SNAP23, VAMP3, and RUSC2 was cloned into pEGFP-C1 to produce the plasmids GFP-STX13, GFP-SNAP23, GFP-VAMP3, and GFP-RUSC2. A dominant-negative mutant, GFP-SNAP23cΔ9, was generated by removing nine residues from the C-terminus of SNAP23. Additional dominant-negative mutants, GFP-VAMP3(cyto) and GFP-STX13(cyto), were generated by constructing truncations that lack the transmembrane regions of the respective

**Fig. 7 | ATG9A differentially regulates unconventional secretion of galectin family members.** Galectin-4-Myc secretion in control (NC) or *ATG9A*-depleted (si*ATG9A*#1) HeLa cells after 24-h DMEM starvation was analyzed by immunoblotting (**a**) and quantified (**b**). Data are presented as mean ± SEM and analyzed using Student's t-test (*n* = 3 biologically independent experiments). **\**p* < 0.01. Galectin-8-Myc secretion in control (NC) or *ATG9A*-depleted (si*ATG9A*#1) HeLa cells after 24-h DMEM starvation was analyzed by immunoblotting (**c**) and quantified (**d**). Data are presented as mean ± SEM and analyzed using Student's t-test (*n* = 3 biologically independent experiments). **\*\**p* < 0.001. GFP-galectin-3 secretion in control (NC) or *ATG9A*-depleted *(si*ATG9A*#1)* HeLa cells following 24-h DMEM starvation was analyzed by immunoblotting (**e**) and quantified (**f**). Data are presented as mean ± SEM and analyzed using Student's t-test (*n* = 3 biologically independent experiments). ns not significant. **g, h** Proteinase K protection assay for galectin-4-Myc in control (Ctrl) and *ATG9A* KO HeLa cells following 24-h DMEM starvation. **g** Immunoblot analysis of post-nuclear supernatant (PNS) and high-speed pellet (HSP) fractions incubated with or without Pro K and Triton X-100. **h** Quantification of protected galectin-4-Myc. Data are presented as mean ± SEM and analyzed using Student's t-test (*n* = 6 biologically independent experiments). **\*\**p* < 0.001. **i, j** Proteinase K

protection assay for galectin-8-Myc in control (Ctrl) and *ATG9A* KO HeLa cells following 24-h DMEM starvation. **i** Immunoblot analysis of PNS and HSP fractions incubated with or without Pro K and Triton X-100. **j** Quantification of protected galectin-8-Myc. Data are presented as mean ± SEM and analyzed using Student's t-test (*n* = 7 biologically independent experiments). **\**p* < 0.01. **k, l** Proteinase K protection assay for GFP-galectin-3 in control (Ctrl) and *ATG9A* KO HeLa cells following 24-h DMEM starvation. **k** Immunoblot analysis of PNS and HSP fractions incubated with or without Pro K and Triton X-100. **l** Quantification of protected GFP-galectin-3. Data are presented as mean ± SEM and analyzed using Student's t-test (*n* = 4 biologically independent experiments). ns not significant. **m, n** HeLa cells expressing GFP-galectin-9 and galectin-8-Myc were subjected to 24-h DMEM starvation, followed by immunostaining for ATG9A and galectin-8-Myc and fluorescence imaging. **m** Representative images. Arrows indicate ATG9A⁺GFP-galectin-9⁺galectin-8-Myc⁺ puncta. Arrowheads indicate ATG9A⁺ galectin-8-Myc⁺ GFP-galectin-9⁻ puncta. Scale bars, 5 μm; insets, 1 μm. **n** Quantification of ATG9A⁺GFP-galectin-9⁺galectin-8-Myc⁺ and ATG9A⁺galectin-8-Myc⁺ puncta. Data are presented as mean ± SEM (*n* = 60 cells examined over 3 independent experiments).

proteins. The following plasmids were generously provided: TMED10-V5, IL-6-FLAG, mIL-1β-FLAG, and FGF2-FLAG by Dr Liang Ge (Tsinghua University); and VAMP3-GFP, RFP-GFP-LC3, GFP-galectin-3, and LAMP1-GFP by Dr Hong Zhang (Institute of Biophysics, Chinese Academy of Sciences). The primer sequences utilized for plasmid construction are presented in Supplementary Table 1.

### Transfection and RNA interference
Transfections with indicated plasmids were carried out using Lipofectamine 2000 (Invitrogen, 11668019) according to the manufacturer's instructions. For RNA interference experiments, cells were transfected with siRNA-Mate (GenePharma, G04003) and harvested 48 h post-transfection. Double-stranded siRNAs were synthesized by JTS Bio, with the oligonucleotide sequences provided in Supplementary Table 2.

### Co-immunoprecipitation, GFP-Trap, and immunoblotting
HeLa cells were transfected with the indicated plasmids for 24 h. Following transfection, cells were collected and lysed in lysis buffer (50 mM Tris-HCl [pH 7.4], 150 mM NaCl, 1 mM EDTA, 1% Triton X-100) supplemented with a protease inhibitor mixture (Selleck, B14001) at 4 °C. The cell lysates were centrifuged at 15,871 × *g* for 15 min, and the resulting supernatant was incubated with diluted primary antibodies and IgG antibodies at 4 °C for 1 h. Subsequently, magnetic beads (MCE, catalog number: HY-K0202) were added to the mixture and incubated for additional 1 h. The magnetic beads were then washed three times with washing buffer (50 mM Tris-HCl [pH 7.4], 150 mM NaCl, 1 mM EDTA, 0.2% Triton X-100) and boiled in 2× SDS-PAGE loading buffer before being used for immunoblotting. For GFP-Trap assays, lysates were immunoprecipitated using GFP-Nanoab-Agarose beads (LABLEAD, HNA-25-500) for 1 h at 4 °C.

For immunoblotting, cells were collected and lysed in lysis buffer (50 mM Tris-HCl [pH 7.4], 150 mM NaCl, 1% NP40, 0.1% SDS) supplemented with a protease inhibitor mixture at 4 °C. After 30 min of incubation, the lysates were centrifuged at 15,871 × *g* for 15 min. Protein concentration was determined using the BCA protein assay (FENGR-BIO, R230518). The supernatant was then mixed with 2× SDS-PAGE loading buffer, boiled, and subjected to immunoblotting. Protein signals were detected using the specified antibodies.

### GST pull-down assay
For the GST pull-down assay, GST or GST-galectin-9 proteins were expressed in *E. coli* BL21 (DE3) cells and purified using glutathione beads (Beyotime, P2253). Bacterial cells were harvested and lysed in lysis buffer (50 mM Tris-HCl [pH 7.4], 150 mM NaCl, 1 mM EDTA, 0.5% NP40, 10% glycerol, and a protease inhibitor mixture) at 4 °C. The lysates were centrifuged at 13,523 × *g* for 30 min at 4 °C, and

the supernatant containing the protein lysate was collected. Subsequently, purified GST or GST-galectin-9 proteins were incubated with glutathione beads and the protein lysate at 4 °C. After incubation, the beads were washed with 1× PBS to remove unbound proteins, Finally, the bound proteins were analyzed by immunoblotting.

### Protein secretion assay
To assess the secretion of galectin-9 or other proteins, HeLa and HEK-293T cells were cultured in DMEM without serum for 12 or 24 h. THP-1 cells were cultured in RPMI-1640 for 24 h. The cell culture medium was then collected and centrifuged at 2000 × *g* for 15 min at 4 °C. The resulting supernatant was concentrated (32-fold) using a 10 kD Amicon filter (Millipore, UFC901096). Simultaneously, cells were lysed in lysis buffer (50 mM Tris-HCl [pH 7.5], 150 mM NaCl, 1 mM EDTA, 1% Triton X-100) supplemented with a protease inhibitor. The lysates were incubated on ice for 30 min and then centrifuged at 15,871 × *g* for 15 min at 4 °C. Immunoblotting analysis was then performed to detect the levels of galectin-9 or other proteins in both the culture medium and cell lysates. For quantification, the secretion levels of galectin-9 or other proteins were normalized to their corresponding total protein levels. Additionally, an LDH assay (Beyotime, C0016) was conducted following the manufacturer's instructions to assess potential cell lysis during the experiment.

### Immunostaining
Cells cultured on coverslips (NEST, 801007) were fixed with 4% paraformaldehyde (PFA) for 20 min at room temperature and permeabilized with 100 μg/ml digitonin (Sigma, D141) for 15 min. Following permeabilization, cells were blocked with 5% goat serum to reduce non-specific binding and then incubated with diluted primary antibodies overnight at 4 °C. The next day, cells were incubated with appropriate secondary antibodies for 1 h at room temperature. Nuclei were counterstained with DAPI, and the coverslips were mounted for visualization. Imaging was performed using a Nikon AX confocal microscope.

### Purification of extracellular vesicles
Extracellular vesicles (EVs) were purified according to previously described methods[30]. Cells were cultured in two 10-cm dishes and starved in DMEM for 24 h. The conditioned medium was collected and subjected to sequential differential centrifugation: 200 × *g* for 10 min, 2000 × *g* for 20 min, 10,000 × *g* for 30 min, and 100,000 × *g* for 120 min (using an NVT100 rotor, Beckman). The resulting EV pellets were resuspended in PBS and ultra-centrifuged at 100,000 × *g* for an additional 70 min. To account for potential seeding differences

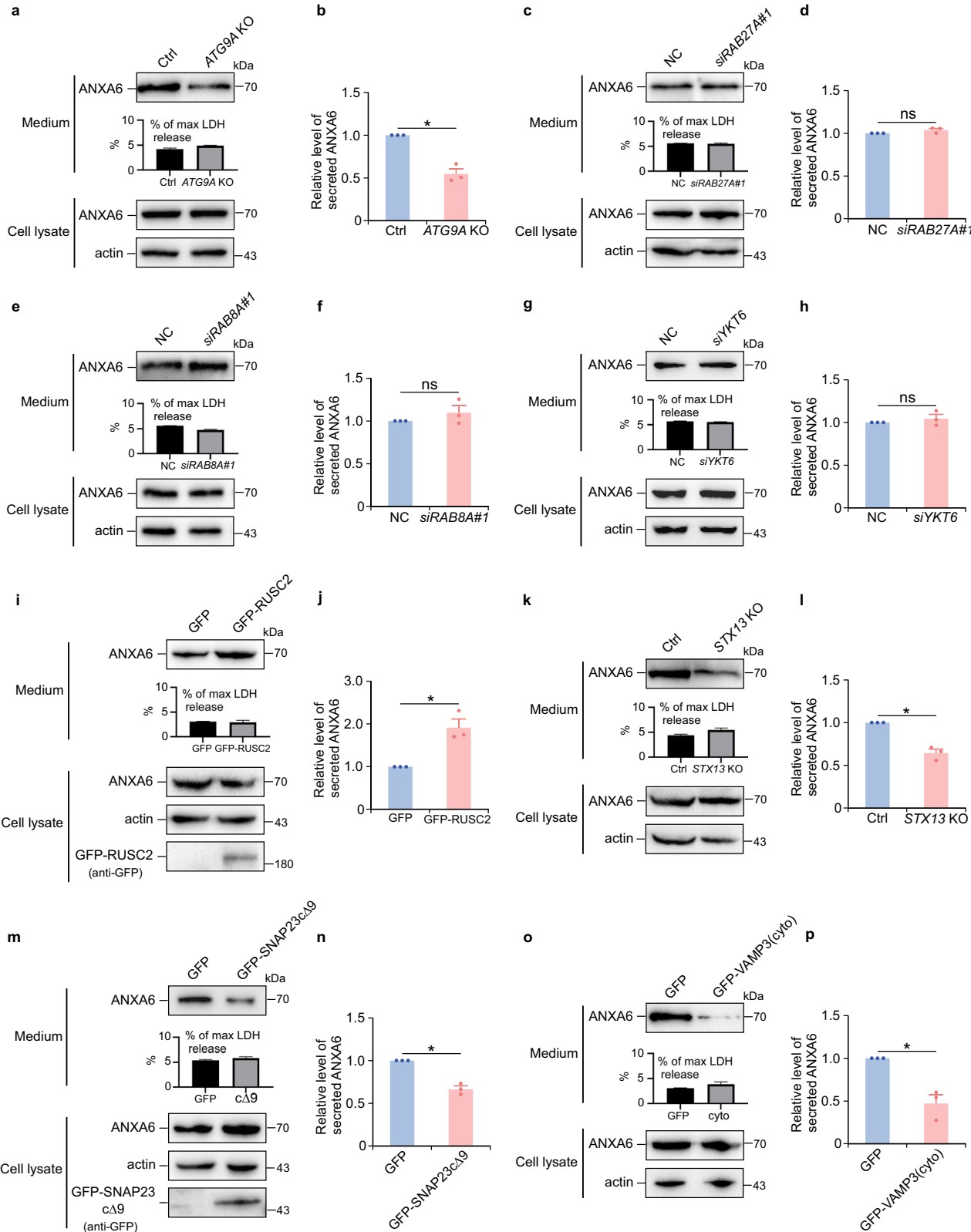

between experimental conditions, EV yields were normalized to whole cell lysate protein concentrations.

## Flow cytometry

Flow cytometry staining for membrane-bound galectin-9 was performed as previously described[54]. Single-cell suspensions were stained with specific primary antibodies or isotype controls as negative controls for 30 min at 4 °C. The following primary antibody was used: goat anti-galectin-9 (AF2045, R&D Systems) at a concentration of 20 µg/ml. Secondary antibody conjugated to Alexa Fluor 555 (Abcam, ab150134) was used at a 1:100 (v/v) dilution. All antibody incubations were carried out in PBS supplemented with 2% fetal bovine serum (FBS).

**Fig. 8 | ATG9A facilitates annexin A6 secretion.** Annexin A6 secretion in control (Ctrl) or *ATG9A* KO HeLa cells after 12-h DMEM starvation was analyzed by immunoblotting (**a**) and quantified (**b**). Data are presented as mean ± SEM and analyzed using Student's t-test (*n* = 3 biologically independent experiments). **p* < 0.05. Annexin A6 secretion in control (NC) or *RAB27A*-depleted (si*RAB27A*#1) HeLa cells after 12-h DMEM starvation was analyzed by immunoblotting (**c**) and quantified (**d**). Data are presented as mean ± SEM and analyzed using Student's t-test (*n* = 3 biologically independent experiments). ns not significant. Annexin A6 secretion in control (NC) or *RAB8A*-depleted (si*RAB8A*#1) HeLa cells after 12-h DMEM starvation was analyzed by immunoblotting (**e**) and quantified (**f**). Data are presented as mean ± SEM and analyzed using Student's t-test (*n* = 3). ns not significant. Annexin A6 secretion in control (NC) or *YKT6*-depleted (si*YKT6*) HeLa cells after 12-h DMEM starvation was analyzed by immunoblotting (**g**) and quantified (**h**). Data are presented as mean ± SEM and analyzed using Student's t-test (*n* = 3 biologically independent experiments). ns not significant. Annexin A6 secretion in GFP- or GFP-RUSC2-expressing HeLa cells after 12-h DMEM starvation was analyzed by immunoblotting (**i**) and quantified (**j**). Data are presented as mean ± SEM and analyzed using Student's t-test (*n* = 3 biologically independent experiments). **p* < 0.05. Annexin A6 secretion in control or *STX13* KO HeLa cells after 12-h DMEM starvation was analyzed by immunoblotting (**k**) and quantified (**l**). Data are presented as mean ± SEM and analyzed using Student's t-test (*n* = 3 biologically independent experiments). **p* < 0.05. Annexin A6 secretion in GFP- or GFP-SNAP23cΔ9-expressing HeLa cells after 12-h DMEM starvation was analyzed by immunoblotting (**m**) and quantified (**n**). Data are presented as mean ± SEM and analyzed using Student's t-test (*n* = 3 biologically independent experiments). **p* < 0.05. Annexin A6 secretion in GFP- or GFP-VAMP3(cyto)-expressing HeLa cells after 12-h DMEM starvation was analyzed by immunoblotting (**o**) and quantified (**p**). Data are presented as mean ± SEM and analyzed using Student's t-test (*n* = 3 biologically independent experiments). * *p* < 0.05.

Flow cytometric analysis was performed using a FACSVerse™ instrument (BD Biosciences) and data were analyzed using FlowJo version X software (Tree Star).

## Proteinase K protection assay
After 24 h of DMEM starvation, cells were collected and resuspended in ice-cold homogenization buffer (0.25 M sucrose, 140 mM NaCl, 1 mM EDTA, 20 mM Tris-HCl [pH 8.0]). Cells were lysed by passing through a 27 G needle ~20 times to ensure effective homogenization. The lysates were then centrifuged at $300 \times g$ for 5 min in a pre-cooled microfuge to obtain the post-nuclear supernatant (PNS). The PNS was further centrifuged at $7600 \times g$ for 5 min at 4 °C to separate the low-speed pellet (LSP) and supernatant. The supernatant was subjected to ultracentrifuged at $100,000 \times g$ for 30 min at 4 °C to isolate the high-speed pellet (HSP) and the high-speed supernatant (HSS). The HSP was resuspended in a homogenization buffer and equally divided into three fractions. One fraction remained untreated, the second fraction was treated with 100 µg/ml protease K (Solarbio, catalog number: p1120), and the third fraction was incubated with both protease K and 0.5% Triton X-100 on ice for 30 min. After treatment, 10% trichloroacetic acid (TCA) was added to all fractions to precipitate proteins. The samples were incubated on ice for 30 min, followed by centrifugation at $21,130 \times g$ for 30 min at 4 °C. The resulting pellets were washed with ice-cold acetone to remove residual TCA, dried to eliminate acetone, and resuspended in 2× SDS-PAGE loading buffer. The resuspended protein samples were subjected to immunoblotting for analysis. To evaluate the membrane protection of galectin-9 in the conditioned medium, the medium was collected after 24 h of DMEM starvation and centrifuged at $2000 \times g$ to remove cellular debris. The resulting supernatant was evenly divided into three fractions and treated with or without proteinase K or 0.5% Triton X-100. The samples were then subjected to TCA protein precipitation and analyzed by immunoblotting.

## Differential centrifugation and membrane fractionation
This assay was performed following a previously described method[38]. HeLa cells were subjected to 24 h of DMEM starvation, after which cells were harvested in B1 buffer (20 mM HEPES-KOH [pH 7.2], 400 mM Sucrose, 1 mM EDTA) supplemented with a protease inhibitor cocktail and 0.3 mM DTT. Subsequently, the cells were homogenized by passing them through a 22 G needle. The resulting homogenate (Total) was processed to a series of centrifugation steps: $3000 \times g$ for 10 min, $25,000 \times g$ for 20 min, and $100,000 \times g$ for 30 min (using an NVT100 rotor, Beckman). The pellets collected from each centrifugation step were boiled in 2× SDS-PAGE loading buffer for further analysis.

## PM damage and PM permeabilization assay
PM damage was induced as previously described[19,44]. Digitonin (Sigma, D141) was diluted in DMEM supplemented with 10% FBS to a final concentration of 150 µg/ml. Cells were treated with 150 µg/ml digitonin for 2 min at 37 °C. After digitonin treatment, cells were incubated with propidium iodide (PI, Solarbio, P8080), diluted in DMEM containing 10% FBS to a final concentration of 100 µg/ml, for 1 min at 37 °C. The cells were then fixed with 4% PFA for 20 min at room temperature. Nuclei were stained with DAPI, and fluorescence was visualized using confocal microscopy (Nikon AX).

## Cell surface biotinylation
Cell surface biotinylation was performed as previously described[19]. Briefly, HeLa cells subjected to the indicated knockdowns were transfected with ATG9A-GFP plasmid and treated with digitonin. After two washes with PBS²⁺ (PBS containing 1 mM MgCl₂ and 0.1 mM CaCl₂), cells were incubated with freshly prepared, pre-cooled 0.4 mM maleimide-PEG2-biotin (MCE, HY-W010764) in PBS (containing 1 mM MgCl₂, 2 mM CaCl₂, and 150 mM NaCl) for 60 min at 4 °C with gentle agitation. To quench unbound maleimide-PEG2-biotin, cells were washed with cold quenching buffer (PBS containing 1 mM MgCl₂, 0.1 mM CaCl₂, and 100 mM glycine) at 4 °C. The cells were then washed twice more with cold PBS²⁺ and collected in ice-cold lysis buffer (containing 50 mM Tris-HCl [pH 8.0], 150 mM NaCl, 1% Triton X-100, and protease inhibitor). After lysis for 30 min, cell lysates were centrifuged at $16,000 \times g$ for 10 min, and protein concentration was measured using the BCA protein assay (FENGRBIO, R230518). Four milligrams of total protein were incubated with 3 µg of GFP antibody overnight at 4 °C. Immunocomplexes were captured using Dynabeads (MCE, Hy K0203), washed three times with PBS, and then boiled with 2× SDS-PAGE loading buffer. Finally, the samples were analyzed by immunoblotting using an HRP-conjugated streptavidin antibody.

## Quantification and statistical analysis
Each experiment was independently repeated at least three times. Statistical analyses were conducted using SPSS and GraphPad Prism 9 software, and data are presented as mean ± SEM. For statistical comparisons, Shapiro-Wilk test of normality was made before parametric analysis. For data following a normal distribution and involving two groups for comparison, a t-test was chosen and applied, with the decision between Student's t-test and Welch's t-test being based on Levene's test for equality of variances. If the variances were found to be equal, Student's t-test was employed; if the variances were unequal, Welch's t-test was used instead. A two-way ANOVA was conducted to compare parameters across multiple groups, after verifying that each group of data had a normal distribution. Subsequently, a Bonferroni multiple comparisons test was performed specifically for the data presented in Fig. 1p. For non-normally distributed data, the Wilcoxon rank sum test was used. Statistically significant was defined as follows: **p* < 0.05, ***p* < 0.01, ****p* < 0.001, indicating increasing levels of significance. *p* > 0.05 was considered not significant (ns).

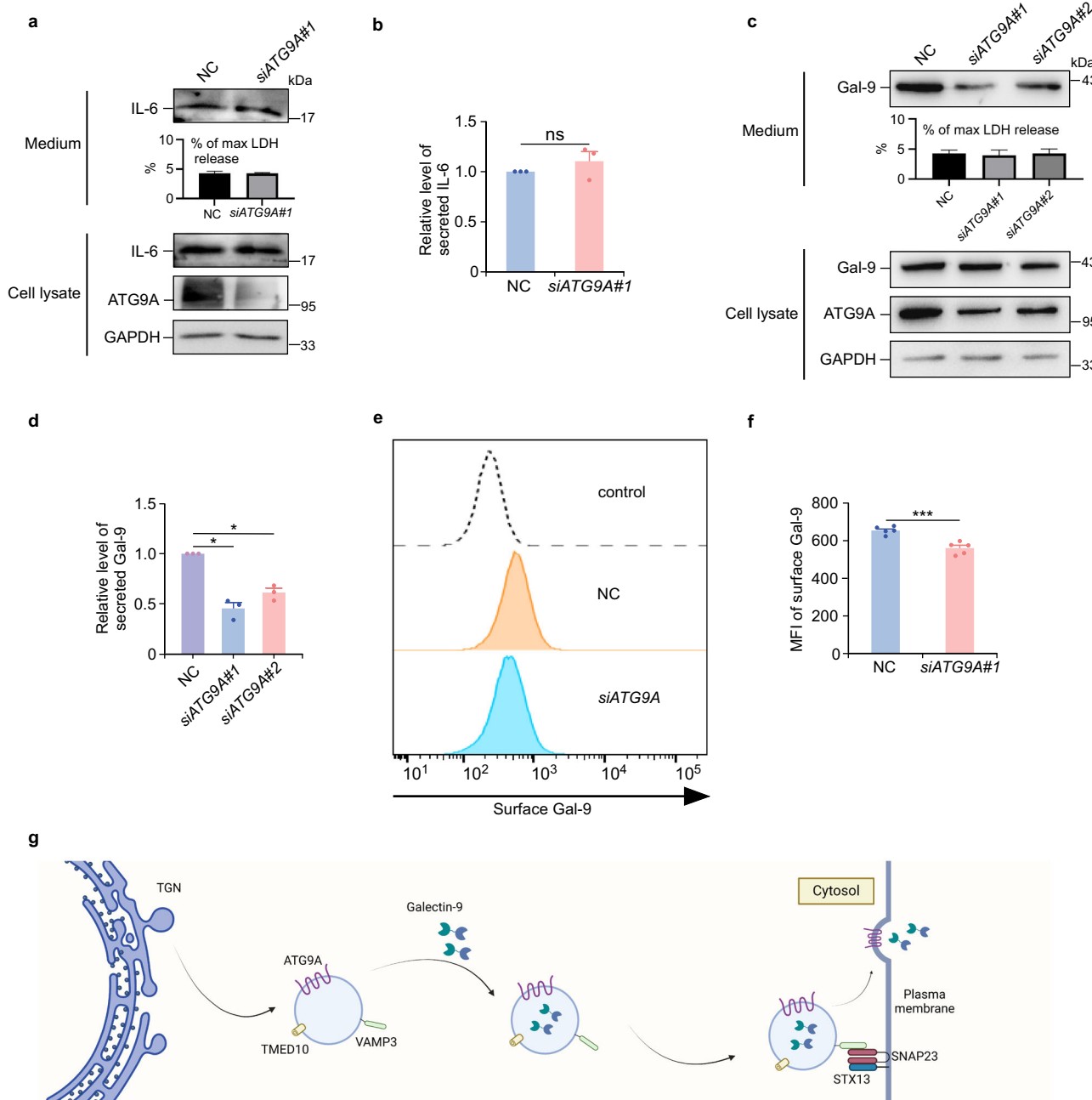

**Fig. 9 | ATG9A regulates galectin-9 secretion and plasma membrane association in THP-1 cells, independent of galectin-9's regulation of IL-6.** IL-6 secretion in control (NC) or *ATG9A*-depleted (si*ATG9A*#1) THP-1 cells following 24-h RPMI-1640 starvation, analyzed by immunoblotting (**a**) and quantified (**b**). Data are presented as mean ± SEM and analyzed using Student's t-test (*n* = 3 biologically independent experiments). ns not significant. Galectin-9 secretion in control (NC) or *ATG9A*-depleted (si*ATG9A*#1 and si*ATG9A*#2) THP-1 cells following 24-h RPMI-1640 starvation, analyzed by immunoblotting (**c**) and quantified (**d**). Data are presented as mean ± SEM and analyzed using Student's t-test (*n* = 3 biologically

independent experiments). *p < 0.05. **e**, **f** Levels of galectin-9 associated with plasma membrane in control (NC) or *ATG9A*-depleted (si*ATG9A*#1) THP-1 cells following 24-h RPMI-1640 starvation, analyzed by flow cytometry (**e**). Numbers below the x-axis mean fluorescence intensity (MFI). **f** Quantification of surface galectin-9 MFI. Data are presented as mean ± SEM and analyzed using Student's t-test (*n* = 5 biologically independent experiments). ***p < 0.001. **g** A model depicting galectin-9 secretion mediated by ATG9 vesicles. This figure was created with BioRender.com with the publication agreement number SN284L52TZ. Created in BioRender. Li, X. (2025) https://BioRender.com/6yt4xys.

---

### Reporting summary

Further information on research design is available in the Nature Portfolio Reporting Summary linked to this article.

### Data availability

All data supporting the findings of this study are presented in the paper and the Supplementary Information. All data are available upon request. Source data are provided with this paper.

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

## Acknowledgements

This work was supported by the National Natural Science Foundation of China (Nos. 32370805 and 31970044 to J.W., No. 32200605 to C.J., No. 82271283 to X.C.). We extend our sincere gratitude Dr Hong Zhang from the Institute of Biophysics, Chinese Academy of Sciences, for providing the VAMP3-GFP, RFP-GFP-LC3, GFP-galectin-3, and LAMP1-GFP plasmids, as well as the *ATG9A* knockout and *FIP200* knockout HeLa cells, and *Atg5* knockout MEF cells. We also thank Dr Liang Ge from Tsinghua University for generously providing the TMED10-V5, IL-6-FLAG, mIL-1β-FLAG, and FGF2-FLAG plasmids.

## Author contributions

J.W. conceived and supervised the study. C.J. and J.W. designed the experiments. W.Z., C.J., X.L., T.H., W.J., Y.L., M.W., and Y.Z. conducted the experiments, supported by X.C., X.W., and X.L. J.L. and C.J. created the graphics. Data analysis was performed by C.J., H.Z., and J.W., and manuscript drafting was done by H.Z. and J.W.

## Competing interests

The authors declare no competing interests.
