## [Transparent Peer Review file · Nature Communications]

Autophagy-independent role of ATG9A vesicles as carriers for galectin-9 secretion

Corresponding Author: Dr Juan Wang

Version 0:

Reviewer comments:

Reviewer #1

(Remarks to the Author)

The authors examine the unconventional secretion of Galectin-9 outside of cells. They propose that ATG9 positive vesicles function as the secretory intermediate and released via a mechanism independent of classical autophagy or secretory autophagy. This model is mainly based on siRNA studies in HeLa cells with some limited extension of results into MEFs and 293T cells. Gal-9 is loaded onto ATG9 vesicles in a TMED10-dependent manner and secreted outside of a cell via the STX13-SNAP23-VAMP3 complex, which mediates the fusion of ATG9 vesicles with the plasma membrane. Although the studies make some interesting observations, it suffers from certain technical shortcomings and conceptual gaps. Notably, the presence of extracellular ATG9 argues against a model in which ATG9 vesicles fuse with the plasma membrane. Overall, additional work is needed to robustly support the conclusions that have been drawn.

- 1) The study makes extensive use of single siRNA targeting oligonucleotides which are prone to off-target effects. Given the unique effects of ATG9 on the secretion of Gal-9, additional siRNA oligos should be employed. The use of ATG9 KO cells is a step, but many other conclusions such as TMED10, RAB8/27, etc, are based on single siRNAs. Furthermore, the main validation of these siRNAs is performed at the mRNA expression level, which is not a robust measure of knockdown efficiency. Additional measures should be analyzed, including immunoblotting for the proteins that are being targeted and in the case of autophagy pathway genes, the effects of these knockdowns on LC3 lipidation (LC3-II) and autophagic flux.
- 2) In Fig 1-O, Gal-9 is being secreted in the 100K fraction, which is not consistent with the author's model that Gal 9 is being released from the plasma membrane in a soluble form. In addition, high levels of ATG9 are observed in the 100K fraction, suggest that a Atg9 positive vesicular intermediate is being released outside of a cell rather than fusing with the plasma membrane. This is a conceptual gap in the proposed model of Gal-9 secretion. The protease protection assays should be performed on extracellular Gal-9 from the conditioned medium to determine whether Gal-9 is secreted as a soluble factor outside of vesicles. Furthermore, the supernatant from the 100K ultracentrifugation fractions should be TCA-precipitated to determine whether soluble Gal-9 is secreted outside of a vesicle-associated intermediate.
- 3) In the studies assessing Gal-9 in TMED10 knockdown cells in Fig 2O, it is unclear from the proteinase K studies whether Gal-9 is residing in a protected intracellular compartment. Although Fig 2P demonstrates a reduction in absolute levels of protected Gal-9 in siTMED10 cells, it is notable that the pre-ProK inputs are also lower in these conditions; hence the degree of protection +/- ProK does not appear to be significant. This parameter should be quantified instead of absolute Gal-9 levels. In addition, IF analysis of Gal-9 and Atg9 co-location in control vs siTMED10 cells may be a complementary approach to ascertain if TMED10 is required for Gal-9 translocation into the Atg9 compartment. As discussed above, additional siRNAs against TMED10 should be employed and confirmation of its translocation activity with known substrates verified.
- 4) The authors also broach a role for secretory autophagy in Gal9 secretion, as both LC3 and ATG5 result in reduced secretion. Curiously, ATG5 is the only LC3 conjugation component that is tested, which makes it difficult to rule out a CASM-like process based on the results shown. Additional LC3-conjugation components (e.g. ATG7 and ATG3) should be tested. On the whole, the relative contributions of autophagy vs. the ATG9-dependent secretory pathway remain unclear.
- 5) The authors should rule out a role for LDELS (Leidal et al. PMID: 31932738), one of the best characterized vesicle-mediated secretory autophagy pathways in ATG9-dependent Gal-9 secretion, especially given the potential contributions of ATG5 and LC3. Does SMPD3 contribute to Gal-9 secretion? Furthermore, does ATG9 knockdown impact the extracellular release of LDELS targets?

Minor:

1) In Fig 1O, what is the isoform of LC3 that is secreted, LC3-I? Curiously, LC3 is primarily secreted in a 25K fraction in these studies, whereas previous studies have demonstrated that extracellular LC3-II is predominantly found in the 100K fraction (e.g. PMID: 31932738). Can the authors provide some explanation for this discrepancy in results?

Reviewer #2

(Remarks to the Author)

Mechanisms of unconventional protein secretion still remain poorly understood. Yet, they concern families of proteins with substantial pathophysiological impact. Zhang and colleagues have analyzed the unconventional secretion process of galectin-9. A role for the autophagy protein ATG9A, the transmembrane P24 trafficking protein 10 (TMED10), and the SNAREs STX13-SNAP23-VAMP3 was identified. It is suggested that galectin-9 enters ATG9A vesicles that upon fusion with the plasma membrane release galectin-9 to the extracellular space. Other unconventional secretion cargoes, i.e., galectin-4, galectin-8, and annexin A6, also are sensitive to the ATG9A mechanism.

The subject area is very timely and of interest to a general readership in cell biology. The experiments are overall convincing and well presented. The manuscript is straight-forward and pleasant to read.

A few points may need further attention to optimize the current submission. These are listed as they appear in the manuscript.

Lines 78-79: It is stated "... suggesting the essential role of ATG9A in galectin-9 secretion". The reduction of galectin-9 secretion that is observed upon ATG9A depletion is quite strong and convincing. However, some secretion remains, which might be due to alternative mechanisms. It might therefore be considered to what extent the use of "essential" might not be a bit too strong. Would "... suggesting a major contribution of ATG9A to galectin-9 secretion" be more appropriate?

Lines 85-86: It is observed that "knockdown of FIP200, ULK1, ATG2A, or ATG2B did not result in a reduction of galectin-9 secretion". What are the positive controls to document that these depletions were effective?

Line 92: It is observed that "knockdown of RAB8A and RAB27A did not reduce galectin-9 secretion". What are the positive controls to document that these depletions were effective?

Lines 97-107: It seems to me that these lines might be better placed after the sentence in lines 78-79. This is just a suggestion. The final call is obviously with the authors.

Lines 108-109: The proteinase K assay is very appropriate. However, one needs to go to the materials and methods section to understand that this assay was performed on post-nuclear supernatants after mechanical lysis of cells. A few words on this in lines 108-109 would help the reader to capture the idea without having to dig too much.

Line 111: From Figure 1N it seems that 100% of cytoplasmic galectin-9 is protected within vesicles. Is that reading of the data correct? If so, this point should be stressed. Galectins have been described to have functions on the cytosolic side of the cytoplasm and in the nucleoplasm, which would be difficult to understand if there's basically no galectin floating freely in the cytosol. Is the same true for galectin-4 and galectin-8 (which are ATG9A sensitive), and not for galectin-3 (which is ATG9A insensitive)?

Line 154: It is concluded that "TMED10 colocalized with ATG9A and galectin-9". This data should be quantified.

Minor points:

- Slight language editing would be optimal.

Reviewer #3

(Remarks to the Author)

In this paper, authors report an interaction between galectin-9 and ATG9A and shed some light into the mechanism(s) by which galectin-9 is secreted to the extracellular matrix. This is an important topic as the pathways that govern galectin transport are still poorly described.

Major points:

1. Colocalisation experiments between galectin-9 and ATG9A show that only 7 % of galectin-9 signal is located in compartments positive for ATG9A. Where does the remaining vast majority of the galectin-9 protein reside then?

2. Similarly, galectin-9 was shown recently to mediate cytokine secretion in a Vamp-3-mediated manner (Santalla Mendez et al., 2023). Given that only a small fraction of galectin-9 is actually embedded into ATG9-containing vesicles, is the mechanism here described by the authors a way of secreting galectin-9 or is gal-9 participating in the secretion of other proteins via vamp-3, SNAP23 vesicles?

3. Authors also determine that galectin-2 and -8 seem to follow similar secretion routes as ATG9A knockdown diminishes

their secretion. Are these galectins co-secreted with galectin-9 or do all galectins travel independently of each other to the extracellular matrix?

4. Galectin secretion is often cell-specific. It would be valuable to discern whether the mechanism described here applies only to epithelial cells or if other cell types (i.e. immune cells) in which galectin-9 is not secreted but remains bound to the extracellular side of the plasma membrane follow the same secretion route.

5. Is the carbohydrate recognition domain important for galectin-9 embedding into ATG9A vesicles or is the process independent of its glycan binding properties?

6. Authors also described that other tandem repeat galectins are secreted in a similar fashion. Does truncation of one of the CRDs (so that gal9 looks more similar to a prototype galectin) prevent its transport via ATG9A?

Minor points:

(i) Efficacy of knockdown of all siRNA transfections (i.e. Ulk1, Rab8a, ATG2A and ATG2B amongst others) should be shown to confirm only ATG9A knockdown decreases gal9 secretion.

(ii) it is not clear from the text why authors focused on ATG9A. What led to investigating ATG9A in relation to galectin-9 secretion?

(iii) Lines 319-328 are not aligned correctly.

(iv) HEK 293 T cells are mentioned in the materials and methods but it is not clear which experiments were conducted with them.

Version 1:

Reviewer comments:

Reviewer #1

(Remarks to the Author)

My previous concerns have been largely addressed in the revision and the rebuttal letter.

Reviewer #2

(Remarks to the Author)

The authors of NCOMMS-24-27965A have responded in full to the comments that I had on the initial version of the manuscript.

Reviewer #3

(Remarks to the Author)

I am overall satisfied with the experiments authors have performed to answer my previous questions/comments. I believe experiments addressing the secretion of other tandem-repeat galectins or galectin-3 are valuable and aid in our understanding of how different galectin subtypes are secreted. Also, all additional quantifications and knockdown validations are solid and add robustness to the manuscript conclusions.

I also appreciate the experiments performed using THP1 as a monocytic cell line to address whether the phenotype the authors saw using HeLa cells was not unique but is potentially conserved across other cell types. However, I would be more concise in the title of the result subsection and in the implications of the data and avoid overstatements not supported by the performed experiments.

"ATG9A-mediated galectin-9 transport is conserved in immune cells" implies the same mechanism is employed by lymphocytes or primary blood myeloid cells but this not something the authors addressed.

Response to Reviewers

Reviewer #1 (Remarks to the Author):

The authors examine the unconventional secretion of Galectin-9 outside of cells. They propose that ATG9 positive vesicles function as the secretory intermediate and released via a mechanism independent of classical autophagy or secretory autophagy,. This model is mainly based on siRNA studies in HeLa cells with some limited extension of results into MEFs and 293T cells. Gal-9 is loaded onto ATG9 vesicles in a TMED10-dependent manner and secreted outside of a cell via the STX13-SNAP23-VAMP3 complex, which mediates the fusion of ATG9 vesicles with the plasma membrane. Although the studies make some interesting observations, the suffers from certain technical shortcomings and conceptual gaps. Notably, the presence of extracellular ATG9 in argues against a model in which ATG9 vesicles fuse with the plasma membrane. Overall, additional work is needed to robustly support the conclusions that have been drawn.

We sincerely thank the reviewer for the time and effort spent evaluating our manuscript and for the constructive feedback. We would like to respectfully clarify that our study does not propose the presence of extracellular ATG9A vesicles. Instead, we demonstrate that intracellular ATG9A-positive vesicles function as carriers that transport cargo proteins within the cell and subsequently fuse with the plasma membrane to mediate the secretion of cargo proteins, such as galectin-9, into the extracellular space.

To further support this model, we have included new data in the revised Figure 2C, which shows that secreted galectin-9 exists in the extracellular environment predominantly in an exposed form with minimal membrane protection. Moreover, ATG9A is undetectable in the culture medium, further supporting our model that ATG9A-positive vesicles mediate protein secretion through membrane fusion rather than vesicle release. We hope this clarification resolves any potential misunderstanding. Additionally, we have revised the relevant sections of the manuscript to ensure that this mechanism is clearly articulated. All specific comments have been thoroughly addressed in the detailed point-by-point responses provided below.

1) The study makes extensive use of single siRNA targeting oligonucleotides which are prone to off-target effects. Given the unique effects of ATG9 on the secretion of Gal-9, additional siRNA oligos should be employed. The use of ATG9 KO cells is a step, but many other conclusions such as TMED10, RAB8/27, etc, are based on single siRNAs. Furthermore, the main validation of these siRNAs is performed at the mRNA expression level, which is not a robust measure of knockdown efficiency. Additional measures should be analyzed, including immunoblotting for the proteins that are being targeted and in the case of autophagy pathway genes, the effects of these knockdowns on LC3 lipidation (LC3-II) and autophagic flux.

We appreciate the reviewer's concerns regarding siRNA validation. We have thoroughly addressed these issues through the following additional experiments:

For *TMED10*, *RAB8A*, and *RAB27A*, we have now incorporated additional siRNA oligonucleotides and validated their efficiencies through immunoblotting of the targeted proteins (Figures 2I–2L, 4E, 4F, S4A–S4D, S7A, S7B). To further confirm the efficiency of these siRNAs, we analyzed their effects on the secretion of known cargo proteins. Consistently, *TMED10* knockdown significantly reduced IL-1 β secretion and its colocalization with LC3 (Figures S7C–S7F). Similarly, knockdown of *RAB8A* and *RAB27A* effectively reduced the secretion of HMGB1 and Annexin A2 (Figure S4E–S4G).

For ATG-related genes, we have also introduced additional siRNA oligonucleotides, validated their knockdown at the protein level through western blotting, and examined their impact on autophagic flux. The results demonstrate effective knockdown and corresponding effects on autophagic flux (Figures 1C–1H, S1C–S1J, S2). Notably, during our validation, we noticed that individual knockdown of *ATG2A* or *ATG2B* had modest effects on autophagic flux. Consequently, we performed dual knockdown of *ATG2A* and *ATG2B*, which effectively disrupted autophagic flux but did not significantly affect galectin-9 secretion (Figures S1I, S1J, S2E–S2H).

2) In Fig 1-O, Gal-9 is being secreted in the 100K fraction, which is not consistent with the author's model that Gal 9 is being released from the plasma membrane in a soluble form. In addition, high levels of ATG9 are observed in the 100K fraction, suggest that a Atg9 positive vesicular intermediate is being released outside of a cell rather than fusing with the plasma membrane. This is a conceptual gap in the proposed model of Gal-9 secretion. The protease protection assays should be performed on extracellular Gal-9 from the conditioned medium to determine whether Gal-9 is secreted as a soluble factor outside of vesicles. Furthermore, the supernatant from the 100K ultracentrifugation fractions should be TCA-precipitated to determine whether soluble Gal-9 is secreted outside of a vesicle-associated intermediate.

We would like to respectfully clarify that the differential centrifugation method employed in our study is designed to separate intracellular components rather than extracellular components. The high levels of ATG9A observed in the 100K pellet reflect cytosolic ATG9A-positive vesicles, not vesicular intermediates being released from the cell. To avoid confusion, we have now added explanatory details to the differential centrifugation section, specifying that it was performed on cell lysates prepared from both control and *ATG9A* knockout HeLa cells.

To address the reviewer's concern regarding extracellular galectin-9, we performed protease protection assays on galectin-9 isolated from the conditioned medium. As shown in the revised Figure 2C, secreted galectin-9 predominantly exists in the extracellular environment in an exposed form with minimal membrane protection. Additionally, ATG9A is undetectable in the culture medium, further supporting our model that ATG9A-positive vesicles mediate protein secretion via plasma membrane fusion rather than vesicle release.

Furthermore, we collected the supernatant from the 100K ultracentrifugation fractions of intracellular components and subjected it to protease protection assays. The results confirmed that the galectin-9 in the 100K supernatant is cytosolic and is not protected by any vesicular structure (Figure S3D).

We hope these additional experiments and clarifications address the reviewer's concerns. The manuscript has been revised accordingly to ensure these points are clearly articulated.

3) In the studies assessing Gal-9 in TMED10 knockdown cells in Fig 2O, it is unclear from the proteinase K studies whether Gal-9 is residing in an protected intracellular compartment. Although Fig 2P demonstrates a reduction in absolute levels of protected Gal-9 in siTMED10 cells, it is notable that the pre-ProK inputs are also lower in these conditions; hence the degree of protection +/- ProK does not appear to be significant. This parameter should be quantified instead of absolute Gal-9 levels. In addition, IF analysis of Gal-9 and Atg9 co-location in control vs siTMED10 cells may be a complementary approach to ascertain if TMED10 is required for Gal-9 translocation into the Atg9 compartment. As discussed above, additional siRNAs against TMED10 should be employed and confirmation of its translocation activity with known substrates verified.

First, regarding the proteinase K protection assays, we performed additional experiments and replaced Figure 2O (now Figure 4I) with representative images showing consistent total levels of galectin-9. Figure 2P (now Figure 4J) has been updated to show the percentage of protected galectin-9 (%). The previous version normalized protected/total Gal-9 relative in control cells to be 1, which caused ambiguity in presentation. This has been clarified and corrected.

Second, we included new immunofluorescence analysis to investigate the colocalization of galectin-9 and ATG9A in control versus si*TMED10* cells. As shown in Figure 4G and 4H, knockdown of *TMED10* resulted in a significant reduction in galectin-9 and ATG9A colocalization.

Third, we validated the specificity and efficiency of *TMED10* knockdown by employing multiple siRNAs. The knockdown efficiency was confirmed through immunoblotting (Figures 4E, 4F, S7A, S7B). We also analyzed the effect of *TMED10* knockdown on IL-1 β secretion. Consistently, *TMED10* knockdown significantly reduced IL-1 β secretion and its colocalization with LC3 (Figure S7C–S7F).

4) The authors also broach a role for secretory autophagy in Gal9 secretion, as both LC3 and ATG5 result in reduced secretion. Curiously, ATG5 is the only LC3 conjugation component that is tested, which makes it difficult to rule out a CASM-like process based on the results shown. Additional LC3-conjugation components (e.g. ATG7 and ATG3) should be tested. On the whole, the relative contributions of autophagy vs. the ATG9-dependent secretory pathway remain unclear.

We have now included data showing that knockdown of another LC3-conjugation component, ATG7, also results in reduced galectin-9 secretion. These findings have been added to Figure S1K and S1L.

We agree with the reviewer that the involvement of LC3-conjugation components in galectin-9 secretion could arise from LC3 conjugation occurring on double-membrane structures, such as secretory autophagosomes, or on single-membrane structures associated with CASM-related pathways. Interestingly, previous studies (PMID: 33201170, 21525242, 33149253, 34597140) have

reported that the CASM pathway does not require ATG9A. To further explore pathway specificity, we examined the LC3-dependent extracellular vesicle loading and secretion (LDELS) pathway, which, similar to CASM, relies on the LC3-conjugation system but is independent of FIP200 and the class III phosphoinositide 3-kinase (PI3K) complex (PMID: 31932738, 36286616). Our results demonstrate that *ATG9A* depletion does not alter the secretion of HNRNPK, a representative LDELS cargo, as shown in Figure 1M and 1N. Both previous studies and our findings suggest that CASM and LDELS pathways depend on LC3 conjugation but are independent of early-stage canonical autophagy regulators, including FIP200, ATG9A, and the PI3K complex.

Treatment with 3-methyladenine (3-MA), an inhibitor of the PI3K complex essential for autophagy but not required for CASM or LDELS (PMID: 33201170, 33149253, 31932738, 36286616), significantly reduced galectin-9 secretion (Figure 1O and 1P). Interestingly, both 3-MA treatment and *ATG9A* knockdown independently reduced galectin-9 secretion by approximately 50%. When 3-MA treatment was combined with *ATG9A* depletion, we observed an additive reduction in galectin-9 secretion (Figure 1O and 1P).

Collectively, based on existing literature and our new data, we believe that the galectin-9 secretion mediated by ATG9A vesicles is likely distinct from CASM or LDELS and that secretory autophagosomes and ATG9A vesicles each account for approximately half of galectin-9 secretion and function through distinct, non-redundant mechanisms.

5) The authors should rule out a role for LDELS (Leidal et al. PMID: 31932738), one of the best characterized vesicle-mediated secretory autophagy pathways in ATG9-dependent Gal-9 secretion, especially given the potential contributions of ATG5 and LC3. Does SMPD3 contribute to Gal-9 secretion? Furthermore, does ATG9 knockdown impact the extracellular release of LDELS targets?

Our results demonstrate that *ATG9A* depletion does not alter the secretion of HNRNPK, a representative LDELS cargo, as shown in revised Figure 1M and 1N. This aligns with previous studies that LDELS pathways depend on LC3 conjugation but are independent of early-stage canonical autophagy regulators, including FIP200 and the PI3K complex (PMID: 31932738, 36286616).

Regarding *SMPD3*, while we observed that its knockdown reduces galectin-9 secretion (Figure Aa and Ab), we also found that *SMPD3* depletion affects the localization and trafficking of ATG9A vesicles. As shown in Figure Ac-Ae, *SMPD3* depletion causes juxtannuclear accumulation of ATG9A and increased colocalization between ATG9A and TFRC, which normally traffics in a sphingomyelin-dependent manner between the plasma membrane and juxtannuclear recycling endosomes. This phenotype closely resembles that observed upon depletion of another sphingomyelin phosphodiesterase, *SMPD1*, which has been shown to regulate ATG9A vesicle trafficking (PMID: 27070082). Based on these observations, we hypothesize that *SMPD3* knockdown may disrupt sphingolipid homeostasis, consequently affecting ATG9A vesicle localization and trafficking, which in turn impacts galectin-9 secretion. We are currently investigating the detailed mechanisms by which *SMPD3* affects ATG9A vesicles and have therefore not included these preliminary findings in the current manuscript.

Figure A. SMPD3 regulates galectin-9 secretion and ATG9A localization.

(a, b) Galectin-9-Myc secretion in control (NC) or *SMPD3*-depleted (siSMPD3#1 and siSMPD3#2) HEK-293T cells following 24-hour DMEM starvation was analyzed by immunoblotting (a) and quantified (b). Data are presented as mean \pm SD (n = 3). **p < 0.01; ***p < 0.001.

(c–e) Immunostaining of ATG9A in control and *SMPD3*-depleted (siSMPD3#2) HeLa cells expressing GFP-TFRC was performed following 24-hour DMEM starvation (c). The percentage of ATG9A puncta colocalized with GFP-TFRC (d) and the proportion of colocalized ATG9A-TFRC puncta in the juxtannuclear region (within 4 μ m from the nucleus) (e) were quantified. Data are presented as mean \pm SD (n = 60 cells per group, three independent experiments). ***p < 0.001.

Minor:

1) In Fig 10, what is the isoform of LC3 that is secreted, LC3-I? Curiously, LC3 is primarily secreted in a 25K fraction in these studies, whereas previous studies have demonstrated that extracellular LC3-II is predominantly found in the 100K fraction (e.g. PMID: 31932738). Can the authors provide some explanation for this discrepancy in results?

We would like to clarify that the centrifugation experiment presented in original Figure 10 (now Figure 2D) was performed on intracellular components, not extracellular fractions. Given that the 25K sediment typically separates membrane structures, and LC3-II is membrane-bound while LC3-

It is diffusely distributed in the cytoplasm without membrane association, we believe that the isoform of LC3 shown in original Figure 1O (now Figure 2D) is LC3-II, which we have now explicitly labeled in Figure 2D. Previous reports also indicate that during the isolation of intracellular components, LC3-II-positive secretory autophagosomes are mainly enriched in the 25K pellet (PMID: 26523392, 27932448).

The reference cited by the reviewer (PMID: 31932738) involves differential centrifugation aimed at isolating extracellular vesicles, where a 10K sediment is used to recover larger extracellular vesicles and a 100K sediment is used for smaller ones. Although both studies utilize differential centrifugation, the differences in the source material (intracellular versus extracellular vesicles) and their distinct properties contribute to variations in their sedimentation patterns during differential centrifugation experiments.

Reviewer #2 (Remarks to the Author):

Mechanisms of unconventional protein secretion still remain poorly understood. Yet, they concern families of proteins with substantial pathophysiological impact. Zhang and colleagues have analyzed the unconventional secretion process of galectin-9. A role for the autophagy protein ATG9A, the transmembrane P24 trafficking protein 10 (TMED10), and the SNAREs STX13-SNAP23-VAMP3 was identified. It is suggested that galectin-9 enters ATG9A vesicles that upon fusion with the plasma membrane release galectin-9 to the extracellular space. Other unconventional secretion cargoes, i.e., galectin-4, galectin-8, and annexin A6, also are sensitive to the ATG9A mechanism.

The subject area is very timely and of interest to a general readership in cell biology. The experiments are overall convincing and well presented. The manuscript is straight-forward and pleasant to read.

A few points may need further attention to optimize the current submission. These are listed as they appear in the manuscript.

We sincerely thank the reviewer for the positive and insightful evaluation of our manuscript. We are delighted that the reviewer found the subject area timely and of interest to a general readership in cell biology. We have carefully addressed all the points raised by the reviewer to improve the quality and impact of our work. Below, we provide detailed responses to each of the comments, along with the corresponding revisions made to the manuscript.

1) Lines 78-79: It is stated "... suggesting the essential role of ATG9A in galectin-9 secretion". The reduction of galectin-9 secretion that is observed upon ATG9A depletion is quite strong and convincing. However, some secretion remains, which might be due to alternative mechanisms. It might therefore be considered to what extent the use of "essential" might not be a bit too strong. Would "... suggesting a major contribution of ATG9A to galectin-9 secretion" be more appropriate?

Thank you for your insightful suggestion. We agree that the use of “essential” may indeed overstate the role of ATG9A in galectin-9 secretion, given the remaining secretion observed upon *ATG9A* depletion. To address this, we have revised the text accordingly to “... suggesting a major contribution of ATG9A to galectin-9 secretion,” as per your recommendation.

2) Lines 85-86: It is observed that “knockdown of FIP200, ULK1, ATG2A, or ATG2B did not result in a reduction of galectin-9 secretion”. What are the positive controls to document that these depletions were effective?

To confirm the effectiveness of these knockdowns, we validated their depletion at the protein level using western blotting and assessed their functional impact on autophagic flux. The results demonstrated successful knockdown of *FIP200*, *ULK1*, *ATG2A*, and *ATG2B*, accompanied by the expected disruptions in autophagic flux. These findings are presented in Figures 1C-1H, S1C-S1J, S2, confirming the effectiveness of the knockdowns.

3) Line 92: It is observed that “knockdown of RAB8A and RAB27A did not reduce galectin-9 secretion”. What are the positive controls to document that these depletions were effective?

To confirm the effectiveness of the *RAB8A* and *RAB27A* knockdowns, we validated their efficiencies through immunoblotting of the targeted proteins (Figures 2I-2L and S4A-S4D). Furthermore, to confirm the specificity and functionality of these siRNAs, we analyzed their effects on the secretion of known cargo proteins. Knockdown of *RAB8A* and *RAB27A* effectively reduced the secretion of HMGB1 and Annexin A2 (Figure S4E-S4G), further demonstrating the effectiveness of these siRNAs.

4) Lines 97-107: It seems to me that these lines might be better placed after the sentence in lines 78-79. This is just a suggestion. The final call is obviously with the authors.

Thank you for your valuable suggestion. To improve the logical flow of the results section, we have restructured the first part accordingly. We now first present the findings showing ATG9A's contribution to galectin-9 secretion. This is followed by the results demonstrating that classic autophagy-related proteins, including FIP200, ULK1, and ATG2A/ATG2B, do not influence galectin-9 secretion. Subsequently, we describe the involvement of the LC3 conjugation machinery, such as LC3, ATG5, and ATG7, in this process. Finally, we address that ATG9A operates independently of secretory autophagy or other LC3 conjugation-dependent pathways, suggesting that ATG9A-mediated galectin-9 secretion is distinct from these known pathways. We believe this restructuring enhances the clarity and logical progression of the results section. We appreciate your suggestion and remain open to making further adjustments if needed.

5) Lines 108-109: The proteinase K assay is very appropriate. However, one needs to go to the materials and methods section to understand that this assay was performed on post-nuclear supernatants after mechanical lysis of cells. A few words on this in lines 108-109 would help the reader to capture the idea without having to dig too much.

Thank you for this suggestion. We have now revised the text to provide essential methodological context for the proteinase K protection assay directly in the results section. We explicitly state that the assay was performed on post-nuclear supernatants following cell lysis, allowing readers to better understand the experimental approach without consulting the methods section. We believe this addition improves the clarity and accessibility of our findings.

6) Line 111: From Figure 1N it seems that 100% of cytoplasmic galectin-9 is protected within vesicles. Is that reading of the data correct? If so, this point should be stressed. Galectins have been described to have functions on the cytosolic side of the cytoplasm and in the nucleoplasm, which would be difficult to understand if there's basically no galectin floating freely in the cytosol. Is the same true for galectin-4 and galectin-8 (which are ATG9A sensitive), and not for galectin-3 (which is ATG9A insensitive)?

Thank you for pointing this out. We apologize for any confusion caused by the original presentation of Figure 1N. In the initial version, we normalized the amount of galectin-9 protected within membrane structures in wild-type cells to 1, which may have led to misunderstanding. In the revised version (Figure 2B), we have updated the y-axis to represent the actual percentage of membrane-protected galectin-9, rather than using normalized values. Our results show that approximately 82% of galectin-9 is protected within membrane structures in wild-type cells, while this protection drops to about 31% in *ATG9A* knockout cells.

Regarding galectin-4 and galectin-8, we have observed that, similar to galectin-9, both are protected within ATG9A-dependent membrane structures. In contrast, galectin-3 exhibits membrane protection that is independent of ATG9A. These findings are now presented in the revised Figure 7G–7L.

7) Line 154: It is concluded that “TMED10 colocalized with ATG9A and galectin-9”. This data should be quantified.

We have quantified the colocalization between TMED10, ATG9A, and galectin-9 in the revised Figure 4C. Our analysis shows that approximately 23% of TMED10 puncta colocalized with ATG9A, and around 14% of TMED10 puncta colocalized with galectin-9.

Minor points:

- Slight language editing would be optimal.

Thank you for your suggestion. We have worked with native English speakers to refine the manuscript and improve the language expression for clarity and readability.

Reviewer #3 (Remarks to the Author):

In this paper, authors report an interaction between galectin-9 and ATG9A and shed some light into the mechanism(s) by which galectin-9 is secreted to the extracellular matrix. This is an important topic as the pathways that govern galectin transport are still poorly described.

We sincerely thank the reviewer for the positive evaluation of our work and for recognizing the importance of studying the pathways governing galectin-9 transport. We are grateful for your constructive comments, which have helped us refine and improve our manuscript. Below, we address your specific remarks in detail.

Major points:

1. Colocalisation experiments between galectin-9 and ATG9A show that only 7 % of galectin-9 signal is located in compartments positive for ATG9A. Where does the remaining vast majority of the galectin-9 protein reside then?

Under normal growth conditions, approximately 7% of galectin-9 signal colocalizes with ATG9A, increasing to about 15% under secretion-inducing conditions (Figure 3B). As shown in Figure B, the majority of galectin-9 localizes to the endosome-lysosome system, with approximately 20% colocalizing with the early endosome marker Rab5, 20% with the late endosome and lysosome marker Rab7, and 20% with the recycling endosome marker TFRC (noting that some overlap may exist between these markers). Additionally, about 22% of galectin-9 puncta colocalize with LC3. This distribution is consistent with previous studies reporting colocalization of galectin-9 with endosomal markers (PMID: 20861448).

Figure B. Galectin-9 predominantly localizes to the endosome-lysosome system

(a, b) Immunostaining of galectin-9 in HeLa cells expressing Rab5-GFP, Rab7-GFP, GFP-TFRC, or immunostained with LC3 following 24-hour starvation in DMEM medium. (a) Arrows indicate galectin-9 puncta colocalized with the indicated markers. Scale bars, 5 μm; insets, 1 μm. (b) Quantification of the percentages of galectin-9 puncta colocalized with the indicated markers. Data are presented as mean ± SEM (n = 60 cells per group from three independent experiments).

The extensive colocalization of galectin-9 with the endosome-lysosome system may be attributed to its diverse established functions. Galectin-9 acts as a sensitive sensor for endosomal escape and endosomal and lysosomal membrane damage (PMID: 26192320, 29625033, 32286269). Furthermore, it interacts with lysosome-associated membrane protein 2 (LAMP2) to maintain lysosome function (PMID: 32855403). In dendritic cells, galectin-9 has been proposed to direct VAMP-3-positive endosomes to the plasma membrane, thereby facilitating cytokine secretion (PMID: 37755527). These findings collectively highlight its versatile roles within the endosome-lysosome system, providing an explanation for its predominant localization in these compartments.

2. Similarly, galectin-9 was shown recently to mediate cytokine secretion in a Vamp-3-mediated manner (Santalla Mendez et al., 2023). Given that only a small fraction of galectin-9 is actually embedded into ATG9-containing vesicles, is the mechanism here described by the authors a way of secreting galectin-9 or is gal-9 participating in the secretion of other proteins via vamp-3, SNAP23 vesicles?

Our data suggests that galectin-9's role in cytokine secretion is distinct from ATG9A-mediated unconventional protein secretion, as supported by several key findings:

First, while *galectin-9* knockdown influences IL-6 secretion as Santalla Mendez et al reported, *ATG9A* depletion does not affect IL-6 secretion in either THP-1 cells or HEK-293T cells (Figures 9A, 9B, S11). This indicates that *ATG9A* serves as a dedicated pathway for unconventional protein secretion, separate from galectin-9's role in cytokine secretion, despite *VAMP3* being important for both processes.

Additionally, we examined whether galectin-9 affects the secretion of other *ATG9A*-dependent cargo proteins. *Galectin-9* knockdown does not impact the unconventional secretion of cargo proteins such as galectin-4 and galectin-8 (Figure C), suggesting that galectin-9 does not participate in the *ATG9A*-mediated secretion of these proteins.

Figure C. Galectin-4 and galectin-8 secretion is not affected by *galectin-9* knockdown

(a, b) Secretion of galectin-4-Myc in control (NC) or *galectin-9*-depleted (siGal-9) HeLa cells following 24-hour starvation in DMEM, analyzed by immunoblotting (a) and quantified (b). Data are presented as mean \pm SEM (n = 3). ns, not significant. (c, d) Secretion of galectin-8-Myc in control (NC) or *galectin-9*-depleted (siGal-9) HeLa cells following 24-hour starvation in DMEM,

analyzed by immunoblotting (c) and quantified (d). Data are presented as mean \pm SEM (n = 3). ns, not significant.

3. Authors also determine that galectin-2 and -8 seem to follow similar secretion routes as ATG9A knockdown diminishes their secretion. Are these galectins co-secreted with galectin-9 or do all galectins travel independently of each other to the extracellular matrix?

We believe the reviewer is referring to galectin-4 and galectin-8. As shown in the revised Figure 7M and 7N, immunofluorescence analysis reveals that galectin-8 displays a punctate distribution, with approximately 20% of its puncta colocalizing with ATG9A but not with galectin-9. Furthermore, only about 6% of the puncta exhibit triple colocalization of galectin-8, galectin-9, and ATG9A. These results suggest that ATG9A may interact independently with galectin-8 and galectin-9, despite some shared trafficking routes.

In contrast, galectin-4 shows a diffuse distribution throughout the cytoplasm (Figure S10; PMID: 20610754, 23824659), with no discernible puncta. This diffuse localization makes it difficult to determine whether galectin-4 is co-transported with galectin-9 via ATG9A-containing vesicles.

Overall, these findings suggest that while tandem galectins may share ATG9A-dependent trafficking mechanisms, they are not necessarily co-transported via the same vesicles.

4. Galectin secretion is often cell-specific. It would be valuable to discern whether the mechanism described here applies only to epithelial cells or if other cell types (i.e. immune cells) in which galectin-9 is not secreted but remains bound to the extracellular side of the plasma membrane follow the same secretion route.

We used flow cytometry to assess the impact of ATG9A on galectin-9 bound to the extracellular side of the plasma membrane in THP-1 cells, a well-established human monocytic cell line widely used to study the biology of monocytes, macrophages, and dendritic cells. In THP-1 cells, galectin-9 exists both as a surface-bound protein on the extracellular side of the plasma membrane and as a secreted factor in the extracellular matrix. Our results show that knockdown of *ATG9A* significantly reduces the levels of galectin-9 on the surface of THP-1 cells (Figure 9E, 9F). This indicates that ATG9A-mediated vesicular transport is crucial for directing galectin-9 to the plasma membrane, regardless of whether it is subsequently secreted into the matrix or remains bound externally.

Furthermore, we found that *ATG9A* knockdown decreased galectin-9 secretion not only in HeLa cells but also in THP-1 and HEK-293T cells (Figures 9C, 9D, S9A, S9B). These findings suggest that the ATG9A-dependent mechanism for galectin-9 secretion is conserved across multiple cell types, including immune cells.

5. Is the carbohydrate recognition domain important for galectin-9 embedding into ATG9A vesicles or is the process independent of its glycan binding properties?.

Both carbohydrate recognition domains (CRDs) of galectin-9 are crucial for ATG9A-mediated secretion. Our experiments demonstrated that the removal of either the N-terminal or C-terminal CRD significantly decreases the dependence of galectin-9 secretion on ATG9A (Figure 3J-3O). This indicates that the glycan binding properties of galectin-9 are important for its incorporation into ATG9A vesicles and subsequent secretion.

6. Authors also described that other tandem repeat galectins are secreted in a similar fashion. Does truncation of one of the CRDs (so that gal9 looks more similar to a prototype galectin) prevent its transport via ATG9A?

As the reviewer anticipated, removing either CRD from galectin-9 results in its secretion becoming independent of ATG9A (Figure 3J-3O). This finding suggests that the unique tandem repeat structure of galectin-9, with its two intact CRDs, is critical for its specific trafficking mechanism via ATG9A vesicles. When galectin-9 is modified to resemble a prototype galectin with only one CRD, its secretion pathway is fundamentally altered, demonstrating the importance of the full CRD configuration in mediating ATG9A-dependent vesicular transport process.

Minor points:

(i) Efficacy of knockdown of all siRNA transfections (i.e Ulk1, Rab8a, ATG2A and ATG2B amongst others) should be shown to confirm only ATG9A knockdown decreases gal9 secretion.

For *TMED10*, *RAB8A*, and *RAB27A*, we have validated their efficiencies through immunoblotting of the targeted proteins (Figures 2I–2L, 4E, 4F, S4A–S4D, S7A, S7B). To further confirm the specificity and efficiency of these siRNAs, we analyzed their effects on the secretion of known cargo proteins. Consistently, *TMED10* knockdown significantly reduced IL-1 β secretion and its colocalization with LC3 (Figure S7C–S7F). Similarly, knockdown of *RAB8A* and *RAB27A* effectively reduced the secretion of HMGB1 and Annexin A2 (Figure S4E–S4G), providing further support for the effectiveness of these siRNAs.

For ATG-related genes, we have also validated their knockdown at the protein level through western blotting, and examined their impact on autophagic flux. The results demonstrate effective knockdown and corresponding effects on autophagic flux (Figures 1C–1H, S1C–S1J, S2). Notably, during our validation, we noticed that individual knockdown of *ATG2A* or *ATG2B* had modest effects on autophagic flux. Consequently, we performed dual knockdown of *ATG2A* and *ATG2B*, which effectively disrupted autophagic flux but did not significantly affect galectin-9 secretion (Figures S1I, S1J, S2E–S2H).

(ii) it is not clear from the text why authors focused on ATG9A. What lead to investigating ATG9A in relation to galectin-9 secretion?

Our investigation of ATG9A arose from analyzing galectin-9 interacting partners. Multiple published mass spectrometry analyses indicated a consistent interaction between galectin-9 and ATG9A (PMID: 31995728, 26186194, 28514442, 31995728). Given that ATG9A vesicles are

highly dynamic, abundant in the cytoplasm, and capable of targeting the plasma membrane, coupled with limited knowledge about their cargo proteins, we explored whether ATG9A vesicles could transport galectin-9. We have now included a concise explanation of this rationale in the Introduction section of our revised manuscript. We appreciate the reviewer for highlighting this opportunity to improve clarity.

(iii) Lines 319-328 are not aligned correctly.

We apologize for the formatting issue in Lines 319–328. This has been corrected, and the text is now properly aligned in the revised manuscript. Thank you for bringing this to our attention.

(iv) HEK 293 T cells are mentioned in the materials and methods but it is not clear which experiments were conducted with them.

We apologize for not clearly indicating the experiments performed with HEK-293T cells in the original manuscript. HEK-293T cells were utilized to confirm our observations in HeLa cells, demonstrating the critical role of the SNARE proteins STX13, SNAP23, and VAMP3 in galectin-9 secretion (Figure S8A–S8F). In addition, *ATG9A* depletion significantly reduced the secretion of galectin-9, galectin-4, and galectin-8, but had no effect on galectin-3 secretion (Figure S9A–S9H). HEK-293T cells were also employed to examine the effects of *ATG9A* and *galectin-9* knockdown on IL-6 secretion. These experiments revealed that *galectin-9* knockdown impairs IL-6 secretion, whereas *ATG9A* knockdown has no impact (Figure S11A and S11B). We have now referenced and integrated these results clearly into the revised manuscript.

REVIEWERS' COMMENTS

Reviewer #3

I also appreciate the experiments performed using THP1 as a monocytic cell line to address whether the phenotype the authors saw using HeLa cells was not unique of but is potentially conserved across other cell types. However, I would be more concise in the title of the result subsection and in the implications of the data and avoid overstatements not supported by the performed experiments.

"ATG9A-mediated galectin-9 transport is conserved in immune cells" implies the same mechanism is employed by lymphocytes or primary blood myeloid cells but this not something the authors addressed.

We appreciate the reviewer's insightful suggestion. We have revised the subtitle of the last section of the results to "ATG9A-mediated galectin-9 transport is conserved in monocytic cells." Additionally, we have made corresponding changes throughout the manuscript, including modifications to the final sentence of the abstract, the last sentence of the results section, and the concluding sentence of the first paragraph of the discussion section to accurately reflect the scope of our findings.